# Faster Differentially Private Top-$k$ Selection:
# A Joint Exponential Mechanism with Pruning

**Hao WU**[*]
University of Waterloo
Canada
`hao.wu1@uwaterloo.ca`

**Hanwen Zhang**
University of Copenhagen
Denmark
`hazh@di.ku.dk`

## Abstract

We study the differentially private top-$k$ selection problem, aiming to identify a sequence of $k$ items with approximately the highest scores from $d$ items. Recent work by Gillenwater et al. (ICML '22) employs a direct sampling approach from the vast collection of $d^{\Theta(k)}$ possible length-$k$ sequences, showing superior empirical accuracy compared to previous pure or approximate differentially private methods. Their algorithm has a time and space complexity of $\tilde{O}(dk)$.

In this paper, we present an improved algorithm with time and space complexity $O(d+k^2/\varepsilon\cdot\ln d)^2$ , where $\varepsilon$ denotes the privacy parameter. Experimental results show that our algorithm runs orders of magnitude faster than their approach, while achieving similar empirical accuracy.

## 1 Introduction

Top-$k$ selection is a fundamental operation with a wide range of applications: *search engines, e-commerce recommendations, data analysis, social media feeds etc*. Here, we consider the setting where the dataset consists of $d$ items evaluated by $n$ people. Each person can cast at most one vote for each item, and vote for unlimited number of items. Our goal is to find a sequence of $k$ items which receives the highest number of votes.

Given that data can contain sensitive personal information such as medical conditions, browsing history, or purchase records, we focus on top-$k$ algorithms that are *differentially private* (Dwork et al., 2006): it is guaranteed that adding/removing an arbitrary single person to/from the dataset does not substantially affect the output. Research for algorithms under this model centers around how accurate the algorithms can be and how efficient they are.

Significant progress has been made in understanding the theoretical boundaries. There are approximate differentially private algorithms (Durfee and Rogers, 2019; Qiao et al., 2021) that achieve asymptotic accuracy lower bound (Bafna and Ullman, 2017; Steinke and Ullman, 2017), and have $O(d)$ time and space usage.

There is also a research endeavor aimed at enhancing the empirical performance of the algorithms. A particularly noteworthy one is the JOINT mechanism by Gillenwater, Joseph, Medina, and Diaz (2022), which exhibits best empirical accuracy across various parameter settings. Diverging from the prevalent *peeling strategy* for top-$k$ selection–wherein items are iteratively selected, removed and repeated $k$ times–the JOINT mechanism considers the sequence holistically, directly selecting an output from the space comprising all $d^{\Theta(k)}$ possible length-$k$ sequences.

While the algorithm has running time and space $\tilde{O}(dk)$, successfully avoiding an exponential time or space consumption, it notably incurs a higher computational cost than its $O(d)$ counterparts. This prompts the interesting question:

---

[*]This work was conducted while the author was a Postdoctoral Fellow at the University of Copenhagen.

[2]A simplified bound from Theorem 4.1 for a wide range of failure probabilities concerning solution quality.

38th Conference on Neural Information Processing Systems (NeurIPS 2024).

*Research Question: Can we design a mechanism equivalent to the* JOINT *mechanism with running time and space linear in* $d$?

**Our Contributions.**   Our paper answers the research question when $k$ is not too large. Specifically,

- We present an improved algorithm with time and space complexity of $O(d + k^2/\varepsilon \cdot \ln d)$

This is an informal statement of Theorem 4.1. When $k \in O(\sqrt{d})$ (a common scenario in practical settings), the time and space complexity simplifies to $\tilde{O}(d)$. Moreover, the proposed algorithm achieves the same asymptotic accuracy guarantee as the JOINT mechanism.

Similar to the JOINT mechanism, our algorithm is an instance of the exponential mechanism (detailed in Section 3) that directly samples from the output space comprising all length-$k$ sequences. We introduce a "group by" sampling framework, which partitions the sequences in the output space into $O(nk)$ subsets, aiming to streamline the sampling process. The framework consists of two steps: sampling a subset and then sampling a sequence from that subset. We provide efficient algorithms for both steps. Furthermore, we introduce a pruning technique to handle outputs with low accuracy uniformly. This technique effectively reduces the number of subsets to $\tilde{O}(k^2)$, leading to an algorithm in $\tilde{O}(d + k^2)$ time and space complexity.

Finally, we perform extensive experiments to

- Verify the theoretical analysis of our algorithm.
- Demonstrate that our algorithm runs 10-100 times faster than JOINT on the tested datasets.
- Show that our algorithm maintains comparable accuracy to JOINT.

**Organization.**   Our paper is structured as follows: Section 2 formally introduces the problem, while Section 3 delves into the necessary preliminaries for our algorithm. Section 4 introduces our novel algorithm, and Section 5 presents our experiment results.

## 2   Problem Description

Let $\mathcal{D} \doteq \{1, \ldots, d\}$ be a set of $d$ items and $\mathcal{U} \doteq \{1, \ldots, n\}$ be a set of $n$ clients. Each client $v \in \mathcal{U}$ can cast at most one vote for each item, and can vote for an unlimited number of items. For each item $i \in \mathcal{D}$, its score $\vec{h}[i]$ is the number of votes it received. The *histogram* is a vector $\vec{h} \doteq (\vec{h}[1], \ldots, \vec{h}[d]) \in [0 \mathbin{..} n]^d$. Define $\mathcal{P}_{\mathcal{D},k} \doteq \left\{ (i_1, \ldots, i_k) \in \mathcal{D}^k : i_1, \ldots i_k \text{ are distinct} \right\}$ be the collection of all possible length-$k$ sequences.

The differentially private top-$k$ selection problem aims at finding a sequence from $\mathcal{P}_{\mathcal{D},k}$ with approximately largest scores, while protecting the privacy of each individual vote.

*Privacy Guarantee.* Two voting histograms $\vec{h}, \vec{h}' \in \mathbb{N}^d$ are neighboring, denoted by $\vec{h} \sim \vec{h}'$, if $\vec{h}'$ can be obtained from $\vec{h}$ by adding or removing an arbitrary individual's votes. Therefore, when $\vec{h} \sim \vec{h}'$, we have $||\vec{h} - \vec{h}'||_\infty \leq 1$, and $\vec{h} \leq \vec{h}'$ or $\vec{h} \geq \vec{h}'$. To protect personal privacy, a top-$k$ selection algorithm should have similar output distributions on neighboring inputs.

**Definition 2.1** (($\varepsilon, \delta$)-Private Algorithm (Dwork and Roth, 2014))**.** *Given* $\varepsilon, \delta > 0$, *a randomized algorithm* $\mathcal{M} : \mathbb{N}^d \to \mathcal{P}_{\mathcal{D},k}$ *is called* ($\varepsilon, \delta$)-*differentially private (DP), if for every* $\vec{h}, \vec{h}' \in \mathbb{N}^d$ *such that* $\vec{h} \sim \vec{h}'$, *and all* $Z \subseteq \mathcal{P}_{\mathcal{D},k}$,

$$\Pr[\mathcal{M}(\vec{h}) \in Z] \leq e^\varepsilon \cdot \Pr[\mathcal{M}(\vec{h}') \in Z] + \delta. \tag{1}$$

*Remark:* An algorithm $\mathcal{M}$ is also called $\varepsilon$-DP for short, if it is ($\varepsilon, 0$)-DP. If an algorithm is $\varepsilon$-DP, it is also called *pure DP*, whereas it is called *approximate DP* if it is ($\varepsilon, \delta$)-DP. Although we present the definition in the context of top-$k$ selection algorithms, it applies more generally to any randomized algorithms $\mathcal{M} : \mathcal{X} \to \mathcal{Y}$, where $\mathcal{X}$ is the input space, which is associated with a symmetric relation $\sim$ that defines neighboring inputs.

# 3 Preliminaries

## 3.1 Exponential Mechanism

The exponential mechanism (McSherry and Talwar, 2007) is a well-known differentially private algorithm for publishing discrete values. Given a general input space $\mathcal{X}$ (associated with a relation $\sim$ which defines neighboring datasets), a finite output space $\mathcal{Y}$, the exponential mechanism $\mathcal{M}_{\mathrm{EXP}} : \mathcal{X} \to \mathcal{Y}$ is a randomized algorithm given by

$$\Pr\left[\mathcal{M}_{\mathrm{EXP}}(x) = y\right] \propto \exp\left(-\varepsilon \cdot \mathcal{E}_{\mathrm{EXP}}(x, y) / (2 \cdot \Delta_{\mathrm{EXP}})\right), \quad \forall x \in \mathcal{X}, \, y \in \mathcal{Y}, \tag{2}$$

where $\mathcal{E}_{\mathrm{EXP}} : \mathcal{X} \times \mathcal{Y} \to \mathbb{R}$ is called the *loss function* measuring how "bad" $y$ is when the input is $x$, and $\Delta_{\mathrm{EXP}}$ is the *sensitivity* of $\mathcal{E}_{\mathrm{EXP}}$ which is the maximum deviation of $\mathcal{E}_{\mathrm{EXP}}$:

$$\Delta_{\mathrm{EXP}} \doteq \max_{x \sim x', y \in \mathcal{Y}} |\mathcal{E}_{\mathrm{EXP}}(x, y) - \mathcal{E}_{\mathrm{EXP}}(x', y)|. \tag{3}$$

**Fact 3.1** (Privacy (McSherry and Talwar, 2007))**.** *The exponential mechanism $\mathcal{M}_{\mathrm{EXP}}$ is $\varepsilon$-DP.*

**Fact 3.2** (Utility Guarantee (McSherry and Talwar, 2007))**.** *For each $\beta \in (0, 1)$, and $\tau \doteq \frac{2 \cdot \Delta_{\mathrm{EXP}}}{\varepsilon} \cdot \ln \frac{|\mathcal{Y}|}{\beta}$, the exponential mechanism $\mathcal{M}_{\mathrm{EXP}}$ satisfies*

$$\Pr\left[\mathcal{E}_{\mathrm{EXP}}(x, \mathcal{M}_{\mathrm{EXP}}(x)) \geq \min_{y \in \mathcal{Y}} \mathcal{E}_{\mathrm{EXP}}(x, y) + \tau\right] \leq \beta, \quad \forall x \in \mathcal{X}.$$

*Implementation.* Given input $x \in \mathcal{X}$, a technique for implementing the exponential mechanism is to add i.i.d. Gumbel noises to the terms of $\{-\varepsilon \cdot \mathcal{E}_{\mathrm{EXP}}(x, y) / (2 \cdot \Delta_{\mathrm{EXP}}) : y \in \mathcal{Y}\}$, and then select the $y$ corresponding to the noisy maximum.

**Definition 3.3.** *Given $b > 0$, the Gumbel distribution with parameter $b$, denoted by $\mathbb{Gumbel}(b)$, has probability density function $p(x) = \frac{1}{b} \cdot \exp\left(-\left(\frac{x}{b} + \exp\left(-\frac{x}{b}\right)\right)\right), \forall x \in \mathbb{R}$.*

**Fact 3.4** ((Yellott, 1977))**.** *Assume that $w_i \geq 0$, for $i \in [m]$, and $X_i \sim \mathbb{Gumbel}(1), i \in [m]$ are independent random variables. Then $\Pr\left[i = \arg\max_{j \in [m]} (X_j + \ln w_j)\right] \propto w_i$.*

It follows that, if $X_y \sim \mathbb{Gumbel}(1), y \in \mathcal{Y}$ are independent random variables, then

$$\Pr\left[y = \arg\max_{y' \in \mathcal{Y}} \{X_{y'} - \varepsilon \cdot \mathcal{E}_{\mathrm{EXP}}(x, y') / (2 \cdot \Delta_{\mathrm{EXP}})\}\right] \propto \exp\left(-\varepsilon \cdot \mathcal{E}_{\mathrm{EXP}}(x, y) / (2 \cdot \Delta_{\mathrm{EXP}})\right).$$

## 3.2 JOINT Mechanism

The JOINT mechanism $\mathcal{M}_{\mathrm{JOINT}} : \mathbb{N}^d \to \mathcal{P}_{\mathcal{D}, k}$ (Gillenwater et al., 2022) is an instance of the exponential mechanism which samples a sequence $\vec{s} = (\vec{s}[1], \ldots, \vec{s}[k])$ directly from $\mathcal{P}_{\mathcal{D}, k}$, with the loss function

$$\mathcal{E}_{\mathrm{JOINT}}(\vec{h}, \vec{s}) \doteq \max_{i \in [k]} \left(\vec{h}_{(i)} - \vec{h}\left[\vec{s}[i]\right]\right), \tag{4}$$

where $\vec{h}_{(i)}$ is the true $i^{(th)}$ largest entry in $\vec{h}$. It can be seen that $\mathcal{E}_{\mathrm{JOINT}}(\cdot)$ has sensitivity $\Delta_{\mathrm{JOINT}} = 1$.

Observe that a naive implementation of this exponential mechanism needs evaluating and storing the scores of $|\mathcal{P}_{\mathcal{D}, k}| = d^{\Omega(k)}$ sequences. Remarkably, Gillenwater, Joseph, Medina, and Diaz (2022) demonstrate that the exponential time and space requirements can be reduced to polynomial.

**Fact 3.5** (JOINT Mechanism (Gillenwater et al., 2022))**.** *There is an implementation of exponential mechanism which directly sample a sequence from $\mathcal{P}_{\mathcal{D}, k}$ according to loss function $\mathcal{E}_{\mathrm{JOINT}}(\vec{h}, \vec{s}) = \max_{i \in [k]} \left(\vec{h}_{(i)} - \vec{h}\left[\vec{s}[i]\right]\right)$ with time $O(dk \log k + d \log d)$ time and space $O(dk)$.*

For completeness, we includes a short proof of Fact 3.5 in Appendix A. Let $\vec{s}^*$ corresponds to the $k$ items with the largest scores. Then clearly $\min_{\vec{s}} \mathcal{E}_{\mathrm{JOINT}}(\vec{h}, \vec{s}) = \mathcal{E}_{\mathrm{JOINT}}(\vec{h}, \vec{s}^*) = 0$. Combining $|\mathcal{P}_{\mathcal{D}, k}| = \binom{d}{k} \cdot k!$ and $\Delta_{\mathrm{JOINT}} = 1$, and applying Fact 3.2, provide the theoretic utility guarantee of JOINT.

**Fact 3.6** (Utility Guarantee)**.** *For each $\beta \in (0, 1)$, $\tau \doteq \left\lceil \frac{2}{\varepsilon} \cdot \ln \frac{\binom{d}{k} \cdot k!}{\beta} \right\rceil \in \Theta\left(\frac{k}{\varepsilon} \cdot (k \ln d + \ln \frac{1}{\beta})\right)$,*

$$\Pr[\mathcal{E}_{\mathrm{JOINT}}(\vec{h}, \mathcal{M}_{\mathrm{JOINT}}(\vec{h})) \geq \tau] \leq \beta. \tag{5}$$

# 4 Algorithm

In this section, we present an algorithm which has similar output distribution as the JOINT mechanism (Gillenwater et al., 2022), but reduces the time and space complexity to $O(d + k \cdot \tau)$. The main result is stated as follows.

**Theorem 4.1.** *Let* $\beta \in (0, 1)$, $\tau \doteq \left\lceil \frac{2}{\varepsilon} \cdot \ln \frac{\binom{d}{k} \cdot k!}{\beta} \right\rceil$, *and* $\mathcal{A} : \mathbb{N}^d \to \mathcal{P}_{\mathcal{D}, k}$ *be the top-k algorithm s.t.*

$$\Pr\left[\mathcal{A}(\vec{h}) = \vec{s}\right] \propto \exp\left(-\varepsilon \cdot \mathcal{E}_{\mathcal{A}}(\vec{h}, \vec{s}) / (2 \cdot \Delta_{\mathcal{A}})\right), \tag{6}$$

*where* $\mathcal{E}_{\mathcal{A}}(\vec{h}, \vec{s}) \doteq \min\left(\mathcal{E}_{\mathrm{JOINT}}(\vec{h}, \vec{s}), \tau\right)$, *and* $\Delta_{\mathcal{A}}$ *is the sensitivity of* $\mathcal{E}_{\mathcal{A}}$. *Then* $\mathcal{A}$ *is* $\varepsilon$-*DP and has an implementation with time and space complexity* $O(d + k \cdot \tau)$. *It satisfies the following condition:*

$$\Pr[\mathcal{E}_{\mathrm{JOINT}}(\vec{h}, \mathcal{A}(\vec{h})) \geq \tau] \leq \beta. \tag{7}$$

*Simplification.* When $1/\beta \in O(d^k)$, the error $\tau$ reduces to $O(d + k^2/\varepsilon \cdot \ln d)$.

It can be verified that the sensitivity $\Delta_{\mathcal{A}} = 1$. Since $\mathcal{A}$ is also an instance of the exponential mechanism, its privacy guarantee naturally follows from this property. As for the utility guarantee, instead of expressing it in terms of $\mathcal{E}_{\mathcal{A}}$ (and directly applying Fact 3.2), we express it in terms of the loss function $\mathcal{E}_{\mathrm{JOINT}}$. The error achieved mirrors that of JOINT, as stated in Corollary 3.6.

*Road map.* In Section 4.1, we first propose a novel "group by" framework for sampling a sequence from the space of all possible length-$k$ sequences. Then, we introduce a new choice of groups to materialize this framework, leading to a new algorithm with $O(d + nk)$ time complexity. In Section 4.2, we propose a new pruning technique that reduces the running time to $O(d + \tau k)$.

A detailed comparison with JOINT is deferred to Section 6, where we will explain JOINT in the context of the novel framework and compare it with our new algorithm.

*To simplify notation, when the input histogram is clear from the context, we use* $\mathcal{E}(\vec{s})$ *as shorthand for* $\mathcal{E}_{\mathrm{JOINT}}(\vec{h}, \vec{s})$, *and* $\mathcal{E}_{\mathcal{A}}(\vec{s})$ *as shorthand for* $\mathcal{E}_{\mathcal{A}}(\vec{h}, \vec{s})$.

## 4.1 Sampling Framework

**Partitioning.** We begin with a novel framework for designing algorithms that produces the same output distribution as JOINT. Let $P_1, \ldots, P_m$ be an arbitrary partition of $\mathcal{P}_{\mathcal{D}, k}$. It is called $\mathcal{E}$-*consistent*, if all sequence belonging the same subset have the same loss (w.r.t loss function $\mathcal{E}$): $\forall i \in [m], \forall \vec{s}, \vec{s}' \in P_i, \mathcal{E}(\vec{s}) = \mathcal{E}(\vec{s}')$. We regard $\mathcal{E}(P_i)$ as the loss of the sequences in $P_i$. Given this partition, we can design an algorithm that reproduces the output distribution of JOINT using a two-step approach:

> **Subset Sampling:** sample a subset $P_i$ with probability proportional to $|P_i| \cdot \exp\left(-\varepsilon \cdot \mathcal{E}(P_i)/2\right)$.
>
> **Sequence Sampling:** sample an $\vec{s} \in P_i$ uniformly.

There can be more than one choices of partitions of $\mathcal{P}_{\mathcal{D}, k}$. We would like to find one which enables efficient sampling algorithms for both steps. We first consider the partition $\mathcal{S}_{r,i}, r \in [0 .. n], i \in [k]$, given by:

$$\mathcal{S}_{r,i} \doteq \left\{ \vec{s} = (\vec{s}[1], \ldots, \vec{s}[k]) \in \mathcal{P}_{\mathcal{D}, k} : \mathcal{E}(\vec{s}) = r \text{ and } \begin{array}{ll} \vec{h}\left[\vec{s}[j]\right] > \vec{h}_{(j)} - r, & \forall j < i \\ \vec{h}\left[\vec{s}[j]\right] = \vec{h}_{(j)} - r, & j = i \\ \vec{h}\left[\vec{s}[j]\right] \geq \vec{h}_{(j)} - r, & \forall j > i \end{array} \right\}. \tag{8}$$

Based on the definition of $\mathcal{E}$ in Equation (4), $\mathcal{S}_{r,i}$ represents the subset of sequences with an loss equal to $r$, and $i$ denotes the index of the first coordinate reaching this loss. Hence, $\mathcal{E}(\mathcal{S}_{r,i}) = r$. Via Fact 3.4, to sample an $\mathcal{S}_{r,i}$ with probability proportional to $|\mathcal{S}_{r,i}| \cdot \exp\left(-\varepsilon \cdot r/2\right)$, we can compute the maximum of $\left\{ X_{r,i} + \ln\left(|\mathcal{S}_{r,i}| \cdot \exp\left(-\varepsilon \cdot r/2\right)\right) : (r, i) \in [0 .. n] \times [k] \right\}$, where $X_{r,i} \sim \mathbb{G}\mathrm{umbel}\,(1)$.

*The first key advantage* of the partition being discussed is that, each $\ln |\mathcal{S}_{r,i}|$ can be expressed as a sum of $k$ terms and it can be computed efficiently.

**Definition 4.2.** *For each* $r \in [0 .. n], j \in [k]$, *define* $C_{r,j} \doteq |\{j' \in [d] : \vec{h}[j'] \geq \vec{h}_{(j)} - r\}|$.

**Lemma 4.3.** *For each $r \in [0 .. n], i \in [k]$, it holds that*

$$\ln |\mathcal{S}_{r,i}| = \sum_{j=1}^{i-1} \ln \left( C_{r-1,j} - (j-1) \right) + \ln \left( C_{r,i} - C_{r-1,i} \right) + \sum_{j=i+1}^{k} \ln \left( C_{r,j} - (j-1) \right). \tag{9}$$

**Lemma 4.4.** *For all $r \in [0 .. n], j \in [k]$, $C_{r,j}$ can be computed in $O(d + nk)$ time. Furthermore, given the $C_{r,j}$'s, for all $r \in [0 .. n]$, $\ln |\mathcal{S}_{r,i}|$ can be computed in $O(nk)$ time.*

The algorithms for proving Lemma 4.4 are detailed in Appendix B. At a high level, for a fixed $r$, the $C_{r,j}, j \in [k]$ constitute a monotone sequence, enabling us to devise a recursion formula to compute them. Additionally, the prefix sums (the first term) and the suffix sums (the last term) in Equation (9) can be pre-computed, simplifying the computation of $\ln |\mathcal{S}_{r,i}|$ to adding only three terms.

Here, we present a proof for Lemma 4.3, offering insights into the structure of $\mathcal{S}_{r,i}$.

*Proof for Lemma 4.3.* It suffices to show that

$$|\mathcal{S}_{r,i}| = \prod_{j=1}^{i-1} \left( C_{r-1,j} - (j-1) \right) \cdot \left( C_{r,i} - C_{r-1,i} \right) \cdot \prod_{j=i+1}^{k} \left( C_{r,j} - (j-1) \right). \tag{10}$$

The proof is via standard counting argument: assume we want to select a sequence $\vec{s} \in \mathcal{S}_{r,i}$. Since $\vec{s}[1] \in \{j' \in [d] : \vec{h}[j'] > \vec{h}_{(1)} - r\}$, the number of possible choices for $\vec{s}[1]$ is

$$|\{j' \in [d] : \vec{h}[j'] > \vec{h}_{(1)} - r\}| = |\{j' \in [d] : \vec{h}[j'] \geq \vec{h}_{(1)} - (r-1)\}| = C_{r-1,1}.$$

The first equality holds because the $\vec{h}[j']$ values are integers.

Next, since $\vec{h}_{(1)} \geq \vec{h}_{(2)}$, it also holds that $\vec{s}[1] \in \{j' \in [d] : \vec{h}[j'] > \vec{h}_{(2)} - r\}$. After determining $\vec{s}[1]$, the number of choices for $\vec{s}[2]$ is $|\{j' \in [d] : \vec{h}[j'] > \vec{h}_{(2)} - r\}| - 1 = C_{r-1,2} - 1$. Continuing this argument, for each $j \in [1 .. i-1]$, the number of choices for $\vec{s}[j]$, after determining $\vec{s}[1 .. j-1]$, is $|\{j' \in [d] : \vec{h}[j'] > \vec{h}_{(j)} - r\}| - (j-1) = C_{r-1,j} - (j-1)$.

Now we consider the number of choices for $\vec{s}[i]$. Since $\vec{s}[1], \ldots, \vec{s}[i-1] \in \{j' \in [d] : \vec{h}[j'] > \vec{h}_{(i)} - r\}$, they do not appear in $\{j' \in [d] : \vec{h}[j'] = \vec{h}_{(i)} - r\}$. The number of choices for $\vec{s}[i]$ is exactly $|\{j' \in [d] : \vec{h}[j'] = \vec{h}_{(i)} - r\}| = C_{r,i} - C_{r-1,i}$.

The cases for $j \in [i+1 .. k]$ are similar to the cases of $j \in [1 .. i-1]$. Since $\vec{h}_{(1)} \geq \vec{h}_{(2)} \geq \cdots \geq \vec{h}_{(j-1)}$, it holds that $\vec{s}[1], \ldots, \vec{s}[j-1] \in \{j' \in [d] : \vec{h}[j'] \geq \vec{h}_{(j)} - r\}$. As a result, for $j \in [i+1 .. k]$, the number of choices for $\vec{s}[j]$, after determining $\vec{s}[1 .. j-1]$, is $C_{r,j} - (j-1)$.

Multiplying the number of choices for each element in $\vec{s} \in \mathcal{S}_{r,i}$, we obtain Equation (10). $\square$

*The second key advantage* of the partition being considered is that, there is an algorithm for sampling a uniform random sequence from $\mathcal{S}_{r,i}$ in $O(d)$ time, as implicitly suggested by the proof for Lemma 4.3. Further details of this implementation are provided in Appendix B.

## 4.2 Pruning

The previous discussion suggests an new algorithm with $O(d + nk)$ running time. Based on Lemma 4.4, computing the $C_{r,j}$ values for $r \in [0 .. n]$ and $j \in [k]$ takes $O(d + nk)$ time. The total time to compute $\ln |\mathcal{S}_{r,i}|$ for $r \in [0 .. n]$ and $i \in [k]$ is $O(nk)$. Finally, sampling a sequence from the chosen $\mathcal{S}_{r,i}$ takes $O(d)$ time. The bottleneck here lies in the $nk$ term, which arises from the need to compute the $C_{r,j}$'s and $\ln |\mathcal{S}_{r,i}|$'s for all $r \in [0 .. n]$.

However, this is unnecessary. We need only to consider the cases for $r \in [0 .. \tau]$. The key observation is that, the probability of sampling an $\vec{s} \in \mathcal{P}_{\mathcal{D},k}$ decreases exponentially with increasing $\mathcal{E}(\vec{s})$.

**Claim 4.5** (Restatement of Fact 3.6). *The probability of sampling an $\vec{s}$ with $\mathcal{E}(\vec{s}) \geq \tau$ is at most $\beta$.*

To provide further insight, we present a short proof here. Let $\mathcal{S}_{\geq \tau} \doteq \{\vec{s} \in \mathcal{P}_{\mathcal{D},k} : \mathcal{E}(\vec{s}) \geq \tau\}$, then

$$\Pr\left[\text{sampling an } \vec{s} \in \mathcal{S}_{\geq \tau}\right] \leq \frac{\sum_{\vec{s} \in \mathcal{S}_{\geq \tau}} e^{-\varepsilon \cdot \mathcal{E}(\vec{s})/2}}{e^{-\varepsilon \cdot \mathcal{E}(\vec{s}^*)/2}} \leq |\mathcal{P}_{\mathcal{D},k}| \cdot e^{-\varepsilon \cdot \tau/2} = \beta, \tag{11}$$

where $\vec{s}^*$ is the $k$ items with the largest scores and $\mathcal{E}(\vec{s}^*) = 0$.

Given this, if we slightly adjust the loss function of sequences in $\mathcal{S}_{\geq \tau}$, their probabilities of being outputted will not be significantly affected. It motivates to consider the truncated loss function: $\mathcal{E}_{\mathcal{A}}(\vec{s}) \doteq \min(\mathcal{E}(\vec{s}), \tau)$, and an algorithm $\mathcal{A}$ which samples an $\vec{s}$ with probability proportional to $e^{-\varepsilon \cdot \mathcal{E}_{\mathcal{A}}(\vec{s})/2}$. As inequality (11) still holds if we the $\mathcal{E}(\cdot)$ with $\mathcal{E}_{\mathcal{A}}(\cdot)$, we immediately obtain the following lemma.

**Lemma 4.6.** *The probability of $\mathcal{A}$ sampling an $\vec{s}$ with $\mathcal{E}(\vec{s}) \geq \tau$ is at most $\beta$.*

*Subset Merging.* The most important benefit of truncated loss is that, it allows us to reduce to the number of subsets in the partition $\mathcal{S}_{r,i}, r \in [0 .. n], i \in [k]$ from $O(nk)$ to $O(\tau k)$. In particular, for each $i \in [k]$, as the sequences in the subsets $\mathcal{S}_{r,i}, r \in [\tau .. n]$ has the same truncated loss, it suffices to merge them into one

$$\mathcal{S}_{\geq \tau, i} \doteq \cup_{r \in [\tau .. n]} \mathcal{S}_{r,i} = \left\{ \vec{s} = (\vec{s}[1], \ldots, \vec{s}[k]) \in \mathcal{P}_{\mathcal{D},k} : \begin{array}{ll} \vec{h}[\vec{s}[j]] > \vec{h}_{(j)} - \tau, & \forall j < i \\ \vec{h}[\vec{s}[j]] \leq \vec{h}_{(j)} - \tau, & j = i \end{array} \right\}. \quad (12)$$

$\mathcal{S}_{\geq \tau, i}$ shares a similar formula on its size as Equation (9) and can be uniformly sampled efficiently.

**Lemma 4.7.** *For each $i \in [k]$, we have*

$$\ln |\mathcal{S}_{\geq \tau, i}| = \sum_{j=1}^{i-1} \ln \left( C_{\tau-1,j} - (j-1) \right) + \ln \left( d - C_{r-1,i} \right) + \sum_{j=i+1}^{k} \ln \left( d - (j-1) \right). \quad (13)$$

**Lemma 4.8.** *Given the $C_{\tau-1,j}$'s, each $\ln |\mathcal{S}_{\geq \tau, i}|$ can be computed in $O(1)$ amortized time.*

The proof for Lemma 4.7 is provided in Appendix A, while an algorithmic proof for Lemma 4.8 can be found in Appendix B.

## 5 Experiment

In this section, we compare our algorithm, referred to as FASTJOINT, with existing state-of-the-art methods on real-world datasets. Our Python implementation is available publicly.[3]

**Datasets.** We utilize six publicly available datasets: Games (Steam video games with purchase counts) (Tamber, 2016), Books (Goodreads books with review counts) (Soumik, 2019), News (Mashable articles with share counts) (Fernandes et al., 2015), Tweets (Tweets with like counts) (Bin Tareaf, 2017), Movies (Movies with rating counts) (Harper and Konstan, 2016) and Foods (Amazon grocery and gourmet foods with review counts) (McAuley et al., 2015). Table 1 summarizes their sizes.

| Dataset | Games | Books | News | Tweets | Movies | Food |
|---------|-------|-------|------|--------|--------|------|
| #items | 5,155 | 11,126 | 39,644 | 52,542 | 59,047 | 166,049 |

Table 1: Dataset Size Summary

**Baselines.** Apart from the JOINT mechanism (Gillenwater et al., 2022), we consider the following two candidates: the peeling variant of permute-and-flip mechanism (McKenna and Sheldon, 2020), denoted PNF-PEEL; and the peeling exponential mechanism (Durfee and Rogers, 2019), denoted CDP-PEEL. We don't compare with other mechanisms, e.g. the Gamma mechanism (Steinke and Ullman, 2016) and the Laplace mechanism (Bhaskar et al., 2010; Qiao et al., 2021), which are empirically dominated by PNF-PEEL and CDP-PEEL respectively (Gillenwater et al., 2022).

PNF-PEEL*:* The permute-and-flip is an $\varepsilon$-DP mechanism for top-1 selection. It can be implemented equivalently by adding exponential noise (with privacy budget $\varepsilon/k$) to each entry of $\vec{h}$ and reporting the item with the highest noisy value (Ding et al., 2021). To report $k$ items, we use the *peeling* strategy: select one item using the mechanism, remove it from the dataset, and repeat this process $k$ times, resulting in a running time of $O(dk)$.

CDP-PEEL*:* The $(\varepsilon, \delta)$-DP peeling exponential mechanism samples $k$ items without replacement, selecting one item at a time using a privacy budget of $\tilde{O}(\varepsilon/\sqrt{k})$ according to the exponential mechanism (McSherry and Talwar, 2007). Durfee and Rogers (2019) demonstrate that CDP-PEEL has an equivalent $O(d)$-time implementation.

The code for all competing algorithms was obtained from publicly accessible GitHub repository by Google Research[4], written in Python.

---
[3] https://github.com/wuhao-wu-jiang/Differentially-Private-Top-k-Selection
[4] https://github.com/google-research/google-research/tree/master/dp_topk

**Experiment Setups.**   The experiments are conducted on macOS system with M2 CPU and 24GB memory. We compare the algorithms in terms of running time and error for different values of $k$, $\varepsilon$ and $\beta$. Note that the parameter $\beta$ (see Theorem 4.1) only affects our algorithm. The $(\varepsilon, \delta)$-DP mechanism CDP-PEEL is configured with a $\delta$ parameter of $10^{-6}$, consistent with prior research (Gillenwater et al., 2022).

*Error Metrics.* We evaluate the quality of a solution $\vec{s}$ using both $\ell_\infty$ and $\ell_1$ errors. The $\ell_\infty$ error is defined as $\max_{i \in [k]} |\vec{h}_{(i)} - \vec{s}[i]|$, while the $\ell_1$ error is given by $\sum_{i \in [k]} |\vec{h}_{(i)} - \vec{s}[i]|$.

*Parameter Ranges.* The parameter ranges tested are as follows:

$$k = 10, 20, \ldots, \underline{100}, \ldots, 200, \qquad \varepsilon = 1/4, 1/2, \underline{1}, 2, 4, \qquad \beta = 2^{-6}, 2^{-8}, \underline{2^{-10}}, 2^{-12}, 2^{-14}.$$

The values indicated by underlining represent the default settings. During experiments where one parameter is varied, the other two parameters are kept at their default values.

## 5.1   Results

All experiments are repeated 200 times. Each figure displays the median running time or $\ell_\infty$ or $\ell_1$ error as the center line, with the shaded region spanning the 25th to the 75th percentiles.

**Varying $k$.**   Figure 1 presents the results for different values $k$. FASTJOINT consistently outperforms JOINT in terms of execution speed, running 10 to 100 times faster across various datasets. FASTJOINT is slower than CDP-PEEL; the later has theoretical time complexity $O(d)$ and therefore this is expected.

We observe "jumps" in running time of JOINT on the *games* and *food* datasets. This phenomenon can also be found in the original work by Gillenwater et al. (2022) in the only running time plot for the *food* dataset. Upon investigation, we found that, as noted in their code comments, the current Python implementation of the *Sequence Sampling* step of JOINT has a worst-case time complexity of $O(dk^2)$ instead of $O(dk)$. Although this step does not constitute a bottleneck in their code and accounts for a constant fraction of the total running time, it still introduces instability in the running time. To delve deeper into this issue, we provide a comparison in the appendix where we plot the running time of JOINT (excluding the *Sequence Sampling* step) against the running time of FASTJOINT (including the *Sequence Sampling* step), resulting in smoother time plots. Even with this adjustment, JOINT remains order of magnitude slower.

Interestingly, for small datasets, FASTJOINT can be slower than PNF-PEEL, which has an $O(dk)$ time complexity. This is because PNF-PEEL has a simple algorithmic structure that can be implemented as $k$ rounds of vector operations: each round involves adding a noisy vector to the input histogram and then selecting an item (not previously selected) with the highest score. It is well-known that vectorized implementations[5] gain significant speed boosts by utilizing dedicated Python numerical libraries such as NumPy (Harris et al., 2020). However, as the dataset size increases (e.g., the food dataset), FASTJOINT outperforms PNF-PEEL in terms of speed.

In terms of solution quality, even with the pruning strategy, FASTJOINT does not experience quality degradation compared to JOINT. It delivers similar performance to JOINT across all datasets and performs particularly well on the *books*, *news*, *tweets*, and *movies* datasets, where there are large gaps between the top-$k$ scores. (Due to space limitations, the complete plots of these score gaps are included in the appendix, with partial plots provided in Figure 2). FASTJOINT consistently outperforms the pure differentially private PNF-PEEL for all values of $k$ and the approximate differentially private CDP-PEEL for at least moderately large $k$. These results align with the findings of Gillenwater et al. (2022), who compared JOINT with PNF-PEEL and CDP-PEEL.

**Varying $\varepsilon$.**   Due to space constraints, Figure 2 presents results for different values of $\varepsilon$ on two typical datasets: one where FASTJOINT performs well and one where it does not. We replaced the $\ell_1$ error plot (as it exhibits similar trends to the $\ell_\infty$ plots) with a plot showing the gap between the top-300 scores[6]. The complete plots across all datasets are included in Appendix C. The running time comparison resembles that of the varying-$k$ plots in Figure 1, with one notable difference: the running time of FASTJOINT exhibits a clear decrease as $\varepsilon$ increases. This observation aligns with our theoretical statement about the running time of FASTJOINT, as detailed in Theorem 4.1.

---

[5]In Appendix B, we discuss in more detail the possibility of implementing FASTJOINT with vertoziation.

[6]The $k$ used for the varying $\varepsilon$ experiments is 100; here we plot the gap for the top-300 scores

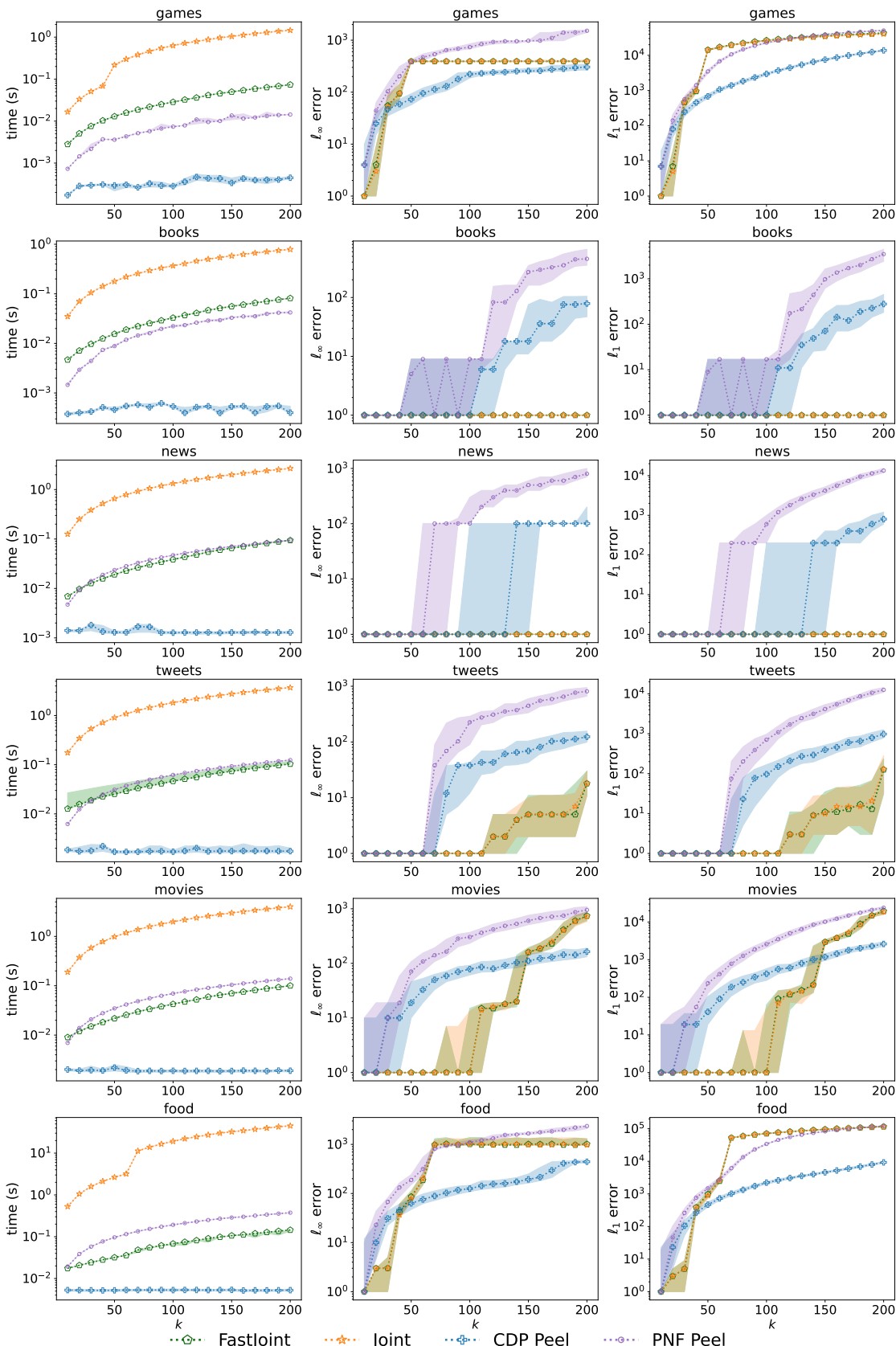

Figure 1: **Left**: Running time vs $k$.  **Center**: $\ell_\infty$ error vs $k$.  **Right**: $\ell_1$ error vs $k$.
The $\ell_1/\ell_\infty$ plots are padded by $1$ to avoid $\log 0$ on the $y$-axis.

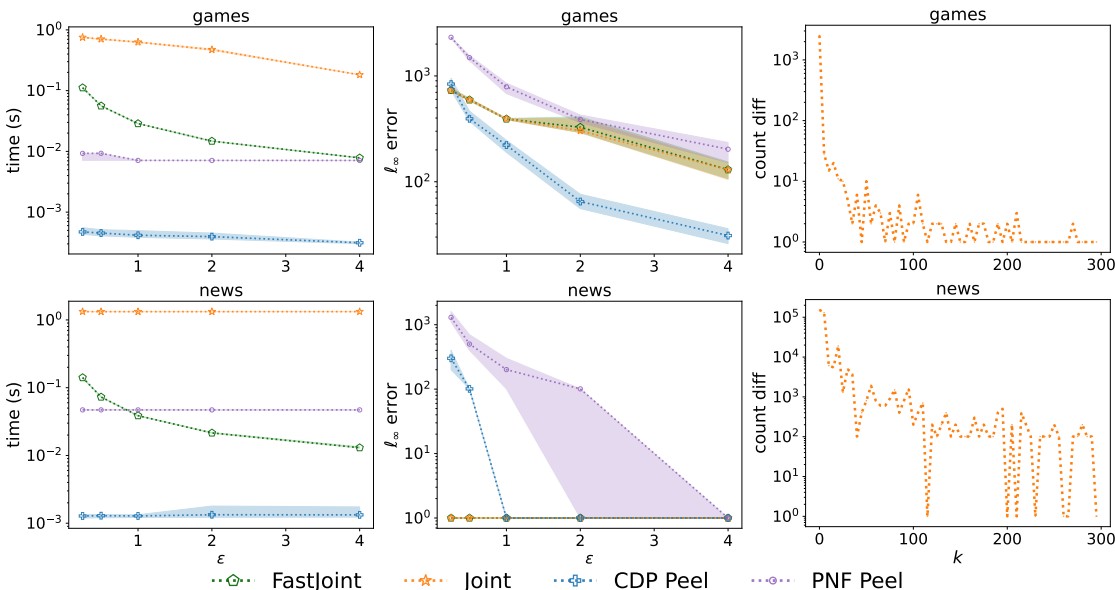

Figure 2: **Left**: Running time Vs $\varepsilon$.   **Center**: $\ell_\infty$ error vs $\varepsilon$.   **Right**: Top-300 scores gaps.
The $\ell_1/\ell_\infty$ plots are padded by $1$ to avoid $\log 0$ on the $y$-axis.

In terms of solution quality, FASTJOINT and JOINT perform particularly well on the *news* dataset and are only slightly better than PNF-PEEL and inferior to CDP-PEEL on the *games* dataset, where the gaps between the large scores in the former dataset are significantly larger than in the latter (note the values on the log-scale y-axis). We provide an informal but informative explanation for this phenomenon: based on Lemma 4.6, FASTJOINT is unlikely to sample sequences with loss greater than $\tau$. Furthermore, when the distribution of top-$k$ score gaps is highly skewed, there are very few sequences with errors between $(0, \tau]$, and the likelihood of sampling these sequences scales with $e^{-O(\varepsilon)}$ as $\varepsilon$ varies. In contrast, the peeling-based mechanism needs to divide its privacy budget by $k$ or $\tilde{O}(\sqrt{k})$ for each round, causing the sampling probability of an error item to scale only with $e^{-O(\varepsilon/k)}$ or $e^{-\tilde{O}(\varepsilon/\sqrt{k})}$, which is higher than $e^{-O(\varepsilon)}$.

**Varying $\beta$.**   Due to space constraints, Figure 3 presents results only for different values of $\beta$ on a medium-sized dataset. Similar plots for other datasets can be found in Appendix C. It is anticipated that JOINT, PNF-PEEL and CDP-PEEL do not exhibit significant performance variation concerning $\beta$. However, it is somewhat surprising that FASTJOINT does not neither. This stability can be attributed to the threshold used for pruning, given by $\tau = \lceil \frac{1}{\varepsilon} \cdot \ln\left(\binom{d}{k} \cdot k!/\beta\right) \rceil$. The numerator inside logarithm term, $\binom{d}{k} \cdot k!$, grows as $d^{\Theta(k)}$, significantly overshadowing $1/\beta$. Consequently, $\tau$ changes only slightly as $\beta$ varies. This experiment demonstrates the robustness of FASTJOINT's pruning strategy concerning the choice of $\beta$.

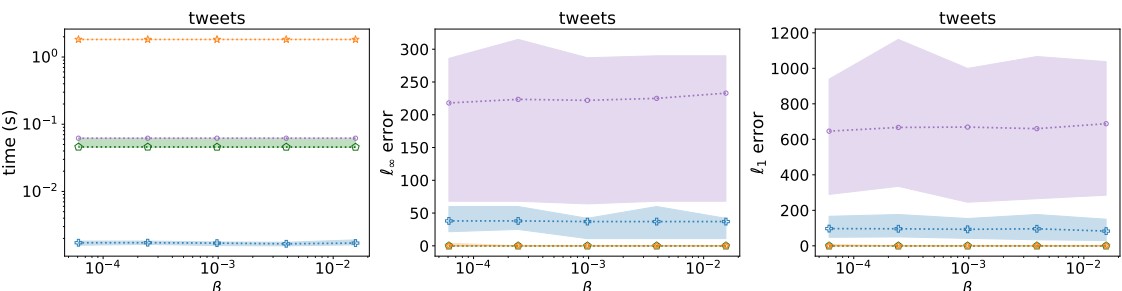

Figure 3: **Left**: Running time vs $\beta$.   **Center**: $\ell_\infty$ error vs $\beta$.   **Right**: $\ell_1$ error vs $\beta$.
The $\ell_1/\ell_\infty$ plots are padded by $1$ to avoid $\log 0$ on the $y$-axis.

# 6 Related Work

**Comparison with JOINT.**   We can also explain JOINT (Gillenwater et al., 2022) within the novel framework proposed in Section 4.1, which consists of the *Subset Sampling* and *Sequence Sampling* steps. Their approach assumes that $\vec{h}[1] \geq \cdots \geq \vec{h}[d]$, which can be achieved by sorting $\vec{h}$. For each $i \in [k]$ and $j \in [d]$, define $\mathcal{E}_{i,j} \doteq \vec{h}[i] - \vec{h}[j]$. The partition they consider is equivalent to the one defined as follows:

$$U_{i,j} \doteq \left\{ \vec{s} = (\vec{s}[1], \ldots, \vec{s}[k]) \in \mathcal{P}_{\mathcal{D},k} : \begin{array}{ll} \vec{h}\big[\vec{s}[\ell]\big] > \vec{h}[\ell] - \mathcal{E}_{i,j}, & \forall \ell < i \\ \vec{s}[\ell] = j, & \ell = i \\ \vec{h}\big[\vec{s}[\ell]\big] \geq \vec{h}[\ell] - \mathcal{E}_{i,j}, & \forall \ell > i \end{array} \right\} , \forall i \in [k], j \in [d].$$

Intuitively, $U_{i,j}$ consists of the length-$k$ sequences, $\vec{s}$, that satisfy: 1) the $i^{\text{th}}$ element in $\vec{s}$ is exactly element $j$; 2) the first $i - 1$ elements in $\vec{s}$ have losses less than $\mathcal{E}_{i,j}$; and 3) the last $k - i$ elements in $\vec{s}$ have losses at most $\mathcal{E}_{i,j}$.

Therefore, the sequences $\vec{s}$ in $U_{i,j}$ share the same loss, $\mathcal{E}_{\text{JOINT}}(\vec{h}, \vec{s}) = \mathcal{E}_{i,j}$ (as defined in Equation (4)), with the first position reaching this loss being the $i^{\text{th}}$ position, where element $j$ appears. Furthermore, sequences in different subsets, $U_{i,j}$ and $U_{i',j'}$, can share the same loss, as it is possible that $\mathcal{E}_{i,j} = \mathcal{E}_{i',j'}$. There are $dk$ subsets in this partition, resulting in a running time of $\tilde{O}(dk)$ for their implementation (Gillenwater et al., 2022).

Applying the pruning technique to this partition can not reduce the number of subsets to $o(dk)$. Let $\tau$ be as defined in Theorem 4.1. For each $i \in [k]$, we aim to find the first $j$ such that $\vec{h}[j] \leq \vec{h}_{(i)} - \tau$. Denote this value by $\sigma(i) \doteq \min\{j \in [d] : \vec{h}[j] \leq \vec{h}_{(i)} - \tau\}$. Following the spirit of our pruning technique, we would like to merge the trailing subsets $U_{i,\sigma(i)}, \ldots, U_{i,d}$ into a single subset $U_{i,\sigma(i)} \doteq \bigcup_{j \geq \sigma(i)} U_{i,j}$ to reduce the number of subsets. However, it is easy to find counterexamples where $\Omega(d)$ items are equal to $\vec{h}_{(i)}$ for all $i \in [k]$. For example, consider the case where $\vec{h}[1] = \vec{h}[2] = \cdots = \vec{h}[d] = c$, for some constant $c$. In such scenarios, we still have $\sigma(i) \in \Omega(d)$, and therefore, in the worst case, the number of subsets remains $\sum_{i \in [k]} \sigma(i) \in \Omega(dk)$.

**Truncated Loss.**   The technique of applying the exponential mechanism with truncated scores was considered by Bhaskar et al. (2010). Their top-$k$ selection algorithm employs a peeling-based approach: it samples $k$ items without replacement, selecting one item at a time using the exponential mechanism with truncated scores. This method is employed because, in their setting, obtaining the scores of the input histogram $\vec{h}$ is expensive, leading them to treat lower-scoring items uniformly by assigning them a small, identical score. In contrast, when all scores of $\vec{h}$ are known, iteratively applying the exponential mechanism to select the top-$k$ items has an equivalent linear time implementation (Durfee and Rogers, 2019) (the CDP-PEEL algorithm in Section 5). Therefore, truncating the scores of $\vec{h}$ is unnecessary in this case.

**Adaptive Private $k$ Selection.**   As the experiments show, the performance of JOINT and FASTJOINT depends on gap size—they perform well when there are large gaps between the top-$k$ items. An orthogonal line of research (Zhu and Wang, 2022) leverages large gaps to privately identify the index $i$ that approximately maximizes the gap between the $i^{\text{th}}$ and $(i + 1)^{\text{th}}$ largest elements. This is followed by testing whether the gap (using techniques like propose-test-release) between the $i^{\text{th}}$ and $(i + 1)^{\text{th}}$ largest elements is sufficiently large, allowing the top-$i$ items to be returned without additional noise. This approach benefits by adding no noise in the final step. However, there are two key differences: 1) it does not guarantee returning at least $k$ items, as $i$ can be less than $k$; 2) more crucially, the top-$i$ items must be returned as an *unordered set*. The first issue can be addressed by iteratively applying the above mechanism. For the second issue, algorithms introduced in this paper, such as CDP-PEEL and FASTJOINT, can serve as subroutines. Notably, FASTJOINT may provide better empirical performance when large gaps exist among the top $i$ items.

**Utility Lower Bound.**   Bafna and Ullman (2017) and Steinke and Ullman (2017) demonstrate that, for approximate private algorithms, existing methods (Durfee and Rogers, 2019; Qiao et al., 2021), including CDP-PEEL (Durfee and Rogers, 2019) as compared in Section 5, achieve theoretically asymptotically optimal privacy-utility trade-offs.

## Acknowledgments and Disclosure of Funding

We thank the anonymous reviewers for their feedback which helped improve the paper. Hao WU was a Postdoctoral Fellow at the University of Copenhagen, supported by Providentia, a Data Science Distinguished Investigator grant from Novo Nordisk Fonden, and affiliated with Basic Algorithms Research Copenhagen (BARC), supported by the VILLUM Foundation grant 16582. Hanwen Zhang is affiliated with Basic Algorithms Research Copenhagen (BARC), supported by the VILLUM Foundation grant 16582. Hanwen Zhang is partially supported by Starting Grant 1054-00032B from the Independent Research Fund Denmark under the Sapere Aude research career programme.

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

# A Missing proofs

*Proof for Fact 3.5.* The proof has been implicitly suggested in our comparison with JOINT in Section 6. We reiterate it here.

We explain JOINT (Gillenwater et al., 2022) within the novel framework proposed in Section 4.1, which consists of the *Subset Sampling* and *Sequence Sampling* steps. Their approach assumes that $\vec{h}[1] \geq \cdots \geq \vec{h}[d]$, which can be achieved by sorting $\vec{h}$. For each $i \in [k]$ and $j \in [d]$, define $\mathcal{E}_{i,j} \doteq \vec{h}[i] - \vec{h}[j]$. The partition they consider is equivalent to the one defined as follows:

$$U_{i,j} \doteq \left\{ \vec{s} = (\vec{s}[1], \ldots, \vec{s}[k]) \in \mathcal{P}_{\mathcal{D},k} : \begin{array}{ll} \vec{h}[\vec{s}[\ell]] > \vec{h}[\ell] - \mathcal{E}_{i,j}, & \forall \ell < i \\ \vec{s}[\ell] = j, & \ell = i \\ \vec{h}[\vec{s}[\ell]] \geq \vec{h}[\ell] - \mathcal{E}_{i,j}, & \forall \ell > i \end{array} \right\}, \forall i \in [k], j \in [d].$$

Intuitively, $U_{i,j}$ consists of the length-$k$ sequences, $\vec{s}$, that satisfy: 1) the $i^{\text{th}}$ element in $\vec{s}$ is exactly element $j$; 2) the first $i-1$ elements in $\vec{s}$ have losses less than $\mathcal{E}_{i,j}$; and 3) the last $k-i$ elements in $\vec{s}$ have losses at most $\mathcal{E}_{i,j}$.

Therefore, the sequences $\vec{s}$ in $U_{i,j}$ share the same loss, $\mathcal{E}_{\text{JOINT}}(\vec{h}, \vec{s}) = \mathcal{E}_{i,j}$ (as defined in Equation (4)), with the first position reaching this loss being the $i^{\text{th}}$ position, where element $j$ appears. Furthermore, sequences in different subsets, $U_{i,j}$ and $U_{i',j'}$, can share the same loss, as it is possible that $\mathcal{E}_{i,j} = \mathcal{E}_{i',j'}$.

Next, we briefly explain how to fulfill the *Subset Sampling* and *Sequence Sampling* steps for the chosen partition.

The *Sequence Sampling* step is straightforward: sampling a sequence uniformly at random from $U_{i,j}$ can be achieved similarly (though not identically) to the approach used in Algorithm 2 for sampling a sequence from $\mathcal{S}_{r,i}$. This process can be completed in $O(d)$ time.

For the *Subset Sampling* step, similar to the algorithm in Section 4.1 and using Fact 3.4, sampling a subset $U_{i,j}$ with probability proportional to $|U_{i,j}| \cdot \exp(-\varepsilon \cdot \mathcal{E}_{i,j}/2)$, can be achieved by computing the maximum of

$$\left\{ X_{i,j} + \ln\left(|U_{i,j}| \cdot \exp(-\varepsilon \cdot \mathcal{E}_{i,j}/2)\right) : (i,j) \in [k] \times [d] \right\},$$

where $X_{i,j} \sim \mathbb{G}\text{umbel}(1)$. The main task is to efficiently compute $\ln|U_{i,j}|$. For each $i \in [k], j \in [d], \ell \in [k]$, we define

$$t_{i,j,>}[\ell] \doteq \left| \left\{ j' \in [d] : \vec{h}[j'] > \vec{h}[\ell] - \mathcal{E}_{i,j} \right\} \right|, \text{ and } t_{i,j,\geq}[\ell] \doteq \left| \left\{ j' \in [d] : \vec{h}[j'] \geq \vec{h}[\ell] - \mathcal{E}_{i,j} \right\} \right|.$$

It can be shown that

$$\ln|U_{i,j}| = \sum_{\ell \in [i-1]} \ln\left(t_{i,j,>}[\ell] - (\ell-1)\right) + \sum_{\ell \in [i+1 .. k]} \ln\left(t_{i,j,\geq}[\ell] - (\ell-1)\right).$$

Furthermore, if we define

$$\tilde{t}_{i,j}[\ell] \doteq \left| \left\{ j' \in [d] : \vec{h}[j'] \geq \vec{h}[\ell] - \mathcal{E}_{i,j} - \left(\frac{i-\ell}{2k} - \frac{j-j'}{2dk}\right) \right\} \right|,$$

we observe that $\left| \frac{i-\ell}{2k} - \frac{j-j'}{2dk} \right| < 1$. When $\ell < i$, $\frac{i-\ell}{2k} - \frac{j-j'}{2dk} \geq \frac{1}{2k} - \frac{d-1}{2dk} > 0$, so $\tilde{t}_{i,j}[\ell] = t_{i,j,>}[\ell]$. On the other hand, when $\ell > i$, $\frac{i-\ell}{2k} - \frac{j-j'}{2dk} \leq -\frac{1}{2k} - \frac{d-1}{2dk} < 0$, so $\tilde{t}_{i,j}[\ell] = t_{i,j,\geq}[\ell]$. Therefore, it follows that

$$\ln|U_{i,j}| = \sum_{\ell \neq i} \ln\left(\tilde{t}_{i,j}[\ell] - (\ell-1)\right).$$

It remains to demonstrate an efficient method for computing $\tilde{t}_{i,j}$. First, we sort the $(i,j) \in [k] \times [d]$ pairs in increasing order based on $\mathcal{E}_{i,j} + \frac{i}{2k} - \frac{j}{2dk}$. For a fixed $i$, the values $\mathcal{E}_{i,j}$ for $j \in [d]$ are already in increasing order, so a sorted sequence of $\mathcal{E}_{i,j} + \frac{i}{2k} - \frac{j}{2dk}$ for $j \in [d]$ can be obtained in $O(d)$ time. We then merge $k$ sorted sequences in $O(dk \log k)$ time using $k$-way merging.

If $(\hat{i}, \hat{j})$ appears immediately after $(i, j)$ in the sorted order, then $\ln |U_{\hat{i},\hat{j}}|$ can be derived from $\ln |U_{i,j}|$ in $O(1)$ time. Specifically, we claim that

$$\tilde{t}_{\hat{i},\hat{j}}[\ell] = \begin{cases} \tilde{t}_{i,j}[\ell], & \text{if } \ell \neq \hat{i}, \\ \tilde{t}_{i,j}[\ell] + 1, & \text{if } \ell = \hat{i}. \end{cases}$$

For the first case, assume for contradiction that there exists some $\ell \neq \hat{i}$ such that $\tilde{t}_{\hat{i},\hat{j}}[\ell] > \tilde{t}_{i,j}[\ell]$. Consequently, there exists $j'$ such that

$$\vec{h}[\ell] - \mathcal{E}_{i,j} - \left( \frac{i-\ell}{2k} - \frac{j-j'}{2dk} \right) > \vec{h}[j'] \geq \vec{h}[\ell] - \mathcal{E}_{\hat{i},\hat{j}} - \left( \frac{\hat{i}-\ell}{2k} - \frac{\hat{j}-j'}{2dk} \right) \tag{14}$$

which implies that

$$\mathcal{E}_{\hat{i},\hat{j}} + \frac{\hat{i}}{2k} - \frac{\hat{j}}{2dk} > \vec{h}[\ell] - \vec{h}[j'] + \left( \frac{\ell}{2k} - \frac{j'}{2dk} \right) = \mathcal{E}_{\ell,j'} + \frac{\ell}{2k} - \frac{j'}{2dk} \geq \mathcal{E}_{i,j} + \frac{i}{2k} - \frac{j}{2dk}. \tag{15}$$

Since the values of $\mathcal{E}_{i,j} + \frac{i}{2k} - \frac{j}{2dk}$ are distinct, it follows that

$$\mathcal{E}_{\hat{i},\hat{j}} + \frac{\hat{i}}{2k} - \frac{\hat{j}}{2dk} > \mathcal{E}_{\ell,j'} + \frac{\ell}{2k} - \frac{j'}{2dk} > \mathcal{E}_{i,j} + \frac{i}{2k} - \frac{j}{2dk}, \tag{16}$$

which contradicts the assumption that $(\hat{i}, \hat{j})$ appears immediately after $(i, j)$ in the sorted order.

For the second case ($\ell = \hat{i}$), it is clear that $\tilde{t}_{\hat{i},\hat{j}}[\ell] \geq \tilde{t}_{i,j}[\ell]$. Additionally, observe that

$$\vec{h}[\hat{i}] - \mathcal{E}_{i,j} - \left( \frac{i-\hat{i}}{2k} - \frac{j-\hat{j}}{2dk} \right) > \vec{h}[\hat{j}] \geq \vec{h}[\hat{i}] - \mathcal{E}_{\hat{i},\hat{j}} - \left( \frac{\hat{i}-\hat{i}}{2k} - \frac{\hat{j}-\hat{j}}{2dk} \right)$$

implying that $\tilde{t}_{\hat{i},\hat{j}}[\ell] \geq \tilde{t}_{i,j}[\ell] + 1$. Finally, we show that it is impossible for $\tilde{t}_{\hat{i},\hat{j}}[\ell]$ to exceed $\tilde{t}_{i,j}[\ell] + 1$. If it did, then, using reasoning similar to Inequalities (14), (15), and (16), we could conclude that there exists some $(\hat{i}, j')$ that appears between $(i, j)$ and $(\hat{i}, \hat{j})$, which leads to a contradiction.

$\square$

*Proof for Lemma 4.7.* The proof is under the same spirit as the proof of Lemma 4.3.

It suffices to show that

$$|\mathcal{S}_{\geq \tau, i}| = \prod_{j=1}^{i-1} \left( C_{\tau-1,j} - (j-1) \right) \cdot \left( d - C_{\tau-1,i} \right) \cdot \prod_{j=i+1}^{k} \left( d - (j-1) \right). \tag{17}$$

Recall Equation (12) that

$$\mathcal{S}_{\geq \tau, i} \doteq \cup_{r \in [\tau \,..\, n]} \mathcal{S}_{r,i} = \left\{ \vec{s} = (\vec{s}[1], \dots, \vec{s}[k]) \in \mathcal{P}_{\mathcal{D},k} : \begin{array}{ll} \vec{h}[\vec{s}[j]] > \vec{h}_{(j)} - \tau, & \forall j < i \\ \vec{h}[\vec{s}[j]] \leq \vec{h}_{(j)} - \tau, & j = i \end{array} \right\}.$$

Assume we want to select a sequence $\vec{s} \in \mathcal{S}_{\geq \tau, i}$. Since $\vec{s}[1] \in \{j' \in [d] : \vec{h}[j'] > \vec{h}_{(1)} - \tau\}$, the number of possible choices for $\vec{s}[1]$ is $|\{j' \in [d] : \vec{h}[j'] > \vec{h}_{(1)} - \tau\}| = |\{j' \in [d] : \vec{h}[j'] \geq \vec{h}_{(1)} - (\tau - 1)\}| = C_{\tau-1,1}$. The first equality holds because the $\vec{h}[j']$ values are integers.

Since $\vec{h}_{(\ell)}$ is non-decreasing, for each $\ell < j < i$, $\vec{h}[\vec{s}[\ell]] > \vec{h}_{(\ell)} - \tau \geq \vec{h}_{(j)} - \tau$, therefore $\vec{h}_{(\ell)} \in \{j' \in [d] : \vec{h}[j'] > \vec{h}_{(j)} - \tau\}$. After determining $\vec{s}[1 \,..\, j-1]$, $\vec{s}[j]$ must be chosen from $\{j' \in [d] : \vec{h}[j'] > \vec{h}_{(j)} - \tau\} \setminus \{\vec{s}[\ell] : \ell < j\}$, so it has $|\{j' \in [d] : \vec{h}[j'] > \vec{h}_{(j)} - \tau\}| - (j-1) = C_{\tau-1,j} - (j-1)$ choices.

Now we consider the number of choices for $\vec{s}[i]$. Since $\vec{s}[1], \dots, \vec{s}[i-1] \in \{j' \in [d] : \vec{h}[j'] > \vec{h}_{(i)} - \tau\}$, they do not appear in $\{j' \in [d] : \vec{h}[j'] \leq \vec{h}_{(i)} - \tau\}$. The number of choices for $\vec{s}[i]$ is exactly $|\{j' \in [d] : \vec{h}[j'] \leq \vec{h}_{(i)} - \tau\}| = d - C_{\tau-1,i}$.

For $j \in [i+1 \,..\, k]$, the number of choices for $\vec{s}[j]$, after determining $\vec{s}[1 \,..\, j-1]$, is $d - (j-1)$.

Multiplying the number of choices for each element in $\vec{s} \in \mathcal{S}_{\geq \tau, i}$, we get

$$|\mathcal{S}_{r,i}| = \prod_{j=1}^{i-1} \left( C_{\tau-1,j} - (j-1) \right) \cdot \left( d - C_{\tau-1,i} \right) \cdot \prod_{j=i+1}^{k} \left( d - (j-1) \right).$$

$\square$

# B   Implementation Details

In this section, we discuss how to implement the **Subset Sampling** and **Sequence Sampling** steps, according to the partition $\{\mathcal{S}_{r,i} : r \in [0\,.\,.\,\tau-1], i \in [k]\} \cup \{\mathcal{S}_{\geq \tau, i} : i \in [k]\}$ induced by the loss $\mathcal{E}_{\mathcal{A}}$.

## B.1   Subset Sampling

The algorithm is in Algorithm 1.

**Computing the $C_{r,j}$.**   Let $f_{\vec{h}} : \mathbb{N} \to 2^{\mathbb{N}}$ be the function given by $f_{\vec{h}}[t] \doteq \{i \in \mathcal{D} : \vec{h}[i] = t\}, \forall t \in \mathbb{N}$. By using standard hash map, $f_{\vec{h}}$ can be computed with $O(d)$ time and space. Based on the definition of the $C_{r,j}$'s and that $\vec{h}$ consists of only integer scores, the following recursion holds,

$$
\begin{aligned}
C_{0,1} &= \left| f_{\vec{h}}\big(\vec{h}_{(1)}\big) \right|, \\
C_{0,j} - C_{0,j-1} &= \left| f_{\vec{h}}\big(\vec{h}_{(j)}\big) \right| \cdot \mathbb{1}_{[\vec{h}_{(j-1)} \neq \vec{h}_{(j)}]}, \quad \forall\, 1 < j \leq k, \\
C_{r,j} - C_{r-1,j} &= \left| f_{\vec{h}}\big(\vec{h}_{(j)} - r\big) \right| \quad\quad\quad\quad\quad \forall\, 1 < r < \tau.
\end{aligned}
\tag{18}
$$

Therefore, $C_{r,j}$'s can be computed in $O(d + \tau k)$ time.

**Computing $\ln |\mathcal{S}_{r,i}|$ and $\ln |\mathcal{S}_{\geq \tau, i}|$.**   To simplify the notation, we apply the following definitions.

**Definition B.1.**

$$
\bar{\mathcal{S}}_{r,i} \doteq \begin{cases} \mathcal{S}_{r,i}, & \text{if } r < \tau \\ \mathcal{S}_{\geq \tau, i}, & \text{if } r = \tau \end{cases}
\tag{19}
$$

$$
\bar{C}_{r,i,} \doteq \begin{cases} C_{r,i}, & \text{if } r < \tau \\ d, & \text{if } r = \tau \end{cases}
\tag{20}
$$

For each $r \in [0\,.\,.\,\tau]$ and each $i \in [1\,.\,.\,k]$, define the prefix and the suffix sums by

$$
\vec{\sigma}[r,i] = \sum_{j=1}^{i} \ln\big(\bar{C}_{r-1,j} - (j-1)\big), \quad \overleftarrow{\sigma}[r,i] = \sum_{j=i}^{k} \ln\big(\bar{C}_{r,j} - (j-1)\big), \quad \forall i \in [k]. \tag{21}
$$

For convenience, we assume $\vec{\sigma}[\cdot, 0] = \overleftarrow{\sigma}[\cdot, k+1] = 0$. Combining Equation (9) and (13), we have

$$
\ln \left| \bar{\mathcal{S}}_{r,i} \right| = \vec{\sigma}[r, i-1] + \ln\big(\bar{C}_{r,i} - \bar{C}_{r-1,i}\big) + \overleftarrow{\sigma}[r, i+1] \tag{22}
$$

A corner case is $r = 0$. By definition, $|\bar{\mathcal{S}}_{0,i}| = 0$ unless $i = 0$. We can set $\bar{C}_{-1,0} = 0$ and the equation still holds.

---

**Algorithm 1** Subset Sampling

---

    **Input:** Histogram $\vec{h}$; Privacy Parameter $\varepsilon$

1: Compute $f_{\vec{h}} : \mathbb{N} \to 2^{\mathbb{N}}$ s.t. $f_{\vec{h}}[t] \doteq \{i \in \mathcal{D} : \vec{h}[i] = t\}, \forall t \in \mathbb{N}$.

2: Compute the $\bar{C}_{r,j}$'s according to Equation (18) and Equation (20)

3: Compute the $\vec{\sigma}$'s and $\overleftarrow{\sigma}$'s according to Equation (21)

4: Compute the $\ln |\bar{\mathcal{S}}_{r,i}|$'s according to Equation (22)

5: Sample $(r, i) \leftarrow \arg\max \left\{ X_{r,i} + \ln\big(|\mathcal{S}_{r,i}| \cdot \exp(-\varepsilon \cdot r / 2)\big) \right\}$, where $X_{r,i} \sim \mathbb{Gumbel}(1)$

6: **return** $(r, i)$

---

## B.2   Sequence Sampling

The algorithm for sequence sampling is Algorithm 2. It follows from the counting argument in Lemma 9 when $r < \tau$ and Lemma 13 when $r = \tau$.

**Lemma B.2.** *Algorithm 2 can be implemented in $O(d)$ time.*

---

**Algorithm 2** Sequence Sampling

**Input:** $(r, i)$
**Ensure:** $\vec{s} \xleftarrow{r} \bar{\mathcal{S}}_{r,i}$
1: Let $\vec{s} \leftarrow \varnothing$ be an empty length-$k$ array
2: **for** $j \leftarrow 1$ to $i-1$ **do**
3:  Sample $\vec{s}[j] \xleftarrow{r} \{\ell \in \mathcal{D} : \vec{h}[\ell] > \vec{h}_{(j)} - r\} \setminus \{\vec{s}[1], \dots, \vec{s}[j-1]\}$

4: Sample an $\vec{s}[j] \xleftarrow{r} \begin{cases} \{\ell \in \mathcal{D} : \vec{h}[\ell] = \vec{h}_{(i)} - r\}, & \text{if } r < \tau \\ \{\ell \in \mathcal{D} : \vec{h}[\ell] \le \vec{h}_{(i)} - \tau\}, & \text{if } r = \tau \end{cases}$

5: **for** $j \leftarrow i+1$ to $k$ **do**

6:  Sample an $\vec{s}[j] \xleftarrow{r} \begin{cases} \{\ell \in \mathcal{D} : \vec{h}[\ell] \ge \vec{h}_{(j)} - r\} \setminus \{\vec{s}[1], \dots, \vec{s}[j-1]\}, & \text{if } r < \tau \\ \mathcal{D} \setminus \{\vec{s}[1], \dots, \vec{s}[j-1]\}, & \text{if } r = \tau \end{cases}$

7: **return** $\vec{s}$

---

*Proof of Lemma B.2.* We discuss different sections of pseudo-codes of Algorithm 2, start by the easy ones.

*Case I : Algorithm 2, line 4.* Clearly, this can be implemented in $O(d)$ time.

*Case II: Algorithm 2, line 5-6, when $r = \tau$.* After the first $i$ entries of $\vec{s}$ are determined, we can create an dynamic array, denoted $\vec{a}$, consisting of elements $\mathcal{D} \setminus \{\vec{s}[1], \dots, \vec{s}[i]\}$. This takes $O(d)$ time. Sampling and removing an item from $\vec{a}$ can be done in $O(1)$ time via standard technique, as described in Algorithm 3 (the $\mathcal{AS}(\cdot)$ procedure).

*Case III: Algorithm 2, line 2-3.* This section can be implemented in $O(d + (k + \tau) \log(k + \tau))$ time, as described in Algorithm 3 (the efficient sequence sampler procedure). It first computes a function $f_{\vec{h}} : \mathbb{N} \to 2^{\mathbb{N}}$, s.t., $f_{\vec{h}}(t) \doteq \{i \in \mathcal{D} : \vec{h}[i] = t\}, \forall t \in \mathbb{N}$. By using standard hash map, $f_{\vec{h}}$ can be computed with $O(d)$ time and space. Then it finds the set $\mathcal{I} \doteq \{t \in \mathbb{N} : f_{\vec{h}}(t) \ne \varnothing \wedge t > \vec{h}_{(k)} - \tau\}$, and store elements in $\mathcal{I}$ as an array. It can be computed in $O(d)$ time. Since an item in $\mathcal{I}$ must equal one of the values of $\vec{h}_{(1)}, \dots, \vec{h}_{(k)}, \vec{h}_{(k)} - 1, \dots, \vec{h}_{(k)} - \tau + 1$, it is easy to see that $|\mathcal{I}| \le k + \tau$. So sorting the items in $\mathcal{I}$ in decreasing order takes $O((k + \tau) \log(k + \tau))$ time. Then the algorithm create an empty dynamic array $\vec{a}$, and a variable POS $= 0$. For each $j \in [i - 1]$, before the sampling step (Algorithm 3, line 18), we claim the following holds:

- POS $= \arg\max_z \mathcal{I}[z] > \vec{h}_{(j)} - r$

- $\vec{a} = \{\ell \in \mathcal{D} : \vec{h}[\ell] > \vec{h}_{(j)} - r\} \setminus \{\vec{s}[1], \dots, \vec{s}[j-1]\}$

This is true for $j = 1$. Now, assume this is true for the $j$ and consider the case for $j + 1$. After the sampling step (Algorithm 3, line 18) in the $j$th iteration, we have POS $= \arg\max_z \mathcal{I}[z] > \vec{h}_{(j)} - r$ and $\vec{a} = \{\ell \in \mathcal{D} : \vec{h}[\ell] > \vec{h}_{(j)} - r\} \setminus \{\vec{s}[1], \dots, \vec{s}[j]\}$. At the $(j+1)$-th iteration, the inner loop (Algorithm 3, lines 15-17) increases POS from $z_1 \doteq \arg\max_z \mathcal{I}[z] > \vec{h}_{(j)} - r)$ to $z_2 \doteq \arg\max_z \mathcal{I}[z] > \vec{h}_{(j+1)} - r$, and expand $\vec{a}$ correspondingly. Since $\{\mathcal{I}[z_1], \dots, \mathcal{I}[z_2]\}$ contains all $t \in [\vec{h}_{(j+1)} - r + 1, \vec{h}_{(j)} - r]$ s.t., $f_{\vec{h}}(t) \ne \varnothing$, after this, $\vec{a}$ becomes

$$\vec{a} = \left\{\ell \in \mathcal{D} : \vec{h}[\ell] > \vec{h}_{(j)} - r\right\} \setminus \{\vec{s}[1], \dots, \vec{s}[j]\} \bigcup \left(\bigcup_{t=\vec{h}_{(j+1)}-r+1}^{\vec{h}_{(j)}-r} f_{\vec{h}}(t)\right)$$

$$= \left\{\ell \in \mathcal{D} : \vec{h}[\ell] > \vec{h}_{(j)} - r\right\} \setminus \{\vec{s}[1], \dots, \vec{s}[j]\} \bigcup \left\{\ell \in [d] : \vec{h}_{(j)} - r \ge \vec{h}[\ell] > \vec{h}_{(j+1)} - r\right\}$$

$$= \left\{\ell \in \mathcal{D} : \vec{h}[\ell] > \vec{h}_{(j+1)} - r\right\} \setminus \{\vec{s}[1], \dots, \vec{s}[j]\}$$

Therefore the invaraints are maintained.

*Case IV: Algorithm 2, line 5-6, when $r < \tau$.* Observe that $\{\ell \in \mathcal{D} : \vec{h}[\ell] \geq \vec{h}_{(j)} - r\} = \{\ell \in \mathcal{D} : \vec{h}[\ell] > \vec{h}_{(j)} - r - 1\}$. Hence we can use similar sampling technique to Case III.

$\square$

---

**Algorithm 3**

---

1: **Procedure** ARRAY SAMPLER $\mathcal{AS}(\vec{a})$
    **Input:** Dynamic array $\vec{a}$
2:    $L \leftarrow$ length of $\vec{a}$
3:    Sample $I \xleftarrow{r} [L]$
4:    Swap $\vec{a}[I]$ and $\vec{a}[L]$
5:    $ans \leftarrow \vec{a}[L]$
6:    Remove $\vec{a}[L]$ from $\vec{a}$
7:    **return** $ans$.

8: **Procedure** EFFICIENT SEQUENCE SAMPLER
    **Input:** Histogram $\vec{h}$; Parameter $i \in [k]$.
9:    Compute the function $f_{\vec{h}} : \mathbb{N} \to 2^{\mathbb{N}}$, s.t., $f_{\vec{h}}(t) \doteq \{\ell \in \mathcal{D} : \vec{h}[\ell] = t\}$, $\forall t \in \mathbb{N}$
10:    Compute $\mathcal{I} \leftarrow \{t \in \mathbb{N} : f_{\vec{h}}(t) \neq \varnothing \wedge t > \vec{h}_{(k)} - \tau\}$ and store it as an array
11:    Sort $\mathcal{I}$ in decreasing order
12:    $\vec{a} \leftarrow$ an empty dynamic array
13:    POS $\leftarrow 0$
14:    **for** $j \leftarrow 1$ to $i - 1$ **do**
15:        **while** POS $<$ length of $\mathcal{I} \wedge \mathcal{I}[\text{POS} + 1] > \vec{h}_{(j)} - r$ **do**
16:            Add the items $f_{\vec{h}}(\mathcal{I}[\text{POS} + 1])$ to the back of $\vec{a}$
17:            POS $\leftarrow$ POS $+ 1$
18:        $\vec{s}[j] \leftarrow \mathcal{AS}(\vec{a})$
19:    **return** $\vec{s}[1], \ldots, \vec{s}[i - 1]$.

---

## B.3 Vectorization

Though FASTJOINT is not implemented yet fully vectorized, we discuss its potential here. Given that FASTJOINT has a runtime of $O(d + k^2/\varepsilon \cdot \ln d)$, the bottleneck lies in the $O(d)$ component for large datasets.

The first $O(d)$ part involves computing the groups $f_{\vec{h}}[t] \doteq \{i \in \mathcal{D} : \vec{h}[i] = t\}$ for each unique value $t$ in $\vec{h}$. This computation could be vectorized using an appropriate library.

The second $O(d)$ component in Sequence Sampling can be eliminated with a careful implementation. Recall that in Algorithm 2, a crucial step is to sample elements uniformly at random from the set

$$\{\ell \in \mathcal{D} : \vec{h}[\ell] > \vec{h}_{(j)} - r\} \setminus \{\vec{s}[1], \ldots, \vec{s}[j - 1]\}, \tag{23}$$

for a possible value of $r \in \{1, 2, \ldots, \tau\}$. Sampling from the set $\{\ell \in \mathcal{D} : \vec{h}[\ell] \geq \vec{h}_{(j)} - r\} \setminus \{\vec{s}[1], \ldots, \vec{s}[j - 1]\}$ can be handled similarly.

To achieve this, we construct an array of at most $k + \tau$ buckets (the cost of constructing this array is covered by the initial $O(d)$ time cost):

$$\left[ f_{\vec{h}}[\vec{h}_{(1)}], f_{\vec{h}}[\vec{h}_{(2)}], \ldots, f_{\vec{h}}[\vec{h}_{(k)}], f_{\vec{h}}[\vec{h}_{(k)} - 1], \ldots, f_{\vec{h}}[\vec{h}_{(k)} - \tau] \right].$$

Assume that each $f_{\vec{h}}[t]$ in this array is itself managed by a dynamic array. Sampling from (23) is then equivalent to sampling uniformly from a prefix of buckets without replacement.

The sampling process involves first selecting a bucket with probability proportional to its size, then drawing an element uniformly at random from that bucket. After sampling, the chosen element is removed from the bucket, which can be managed efficiently using a dynamic array. This approach removes the dependency on $d$ in the sampling step.

## C  Supplementary Plots

In this section, we provide supplementary plots for our experiments:

- Figure 4 illustrates the gaps between large-score items for all tested datasets.
- Figure 5 displays the algorithm's running time, $\ell_\infty$ error, and $\ell_1$ error versus $\epsilon$.
- Figure 6 showcases the algorithm's running time, $\ell_\infty$ error, and $\ell_1$ error versus $\beta$.
- Figure 7 depicts the running time of JOINT (excluding time from the Sequence Sampling step) versus the running time of our proposed algorithm FASTJOINT (including time from the Sequence Sampling step), over all tested datasets. Given this, our algorithm still runs orders of magnitude faster than JOINT. Due to time constraints, we only repeated the experiments 5 times to generate the plots. This is acceptable since, according to the previous experiments, the running time of the algorithms is quite stable.

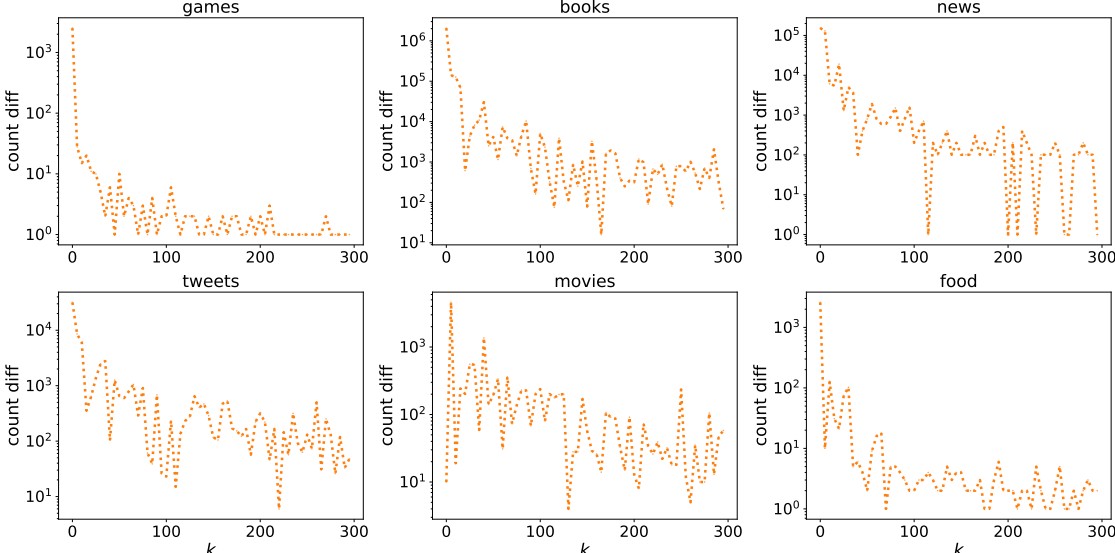

Figure 4: The gaps between the top-$k$ scores (for $k = 300$) for all tested datasets.

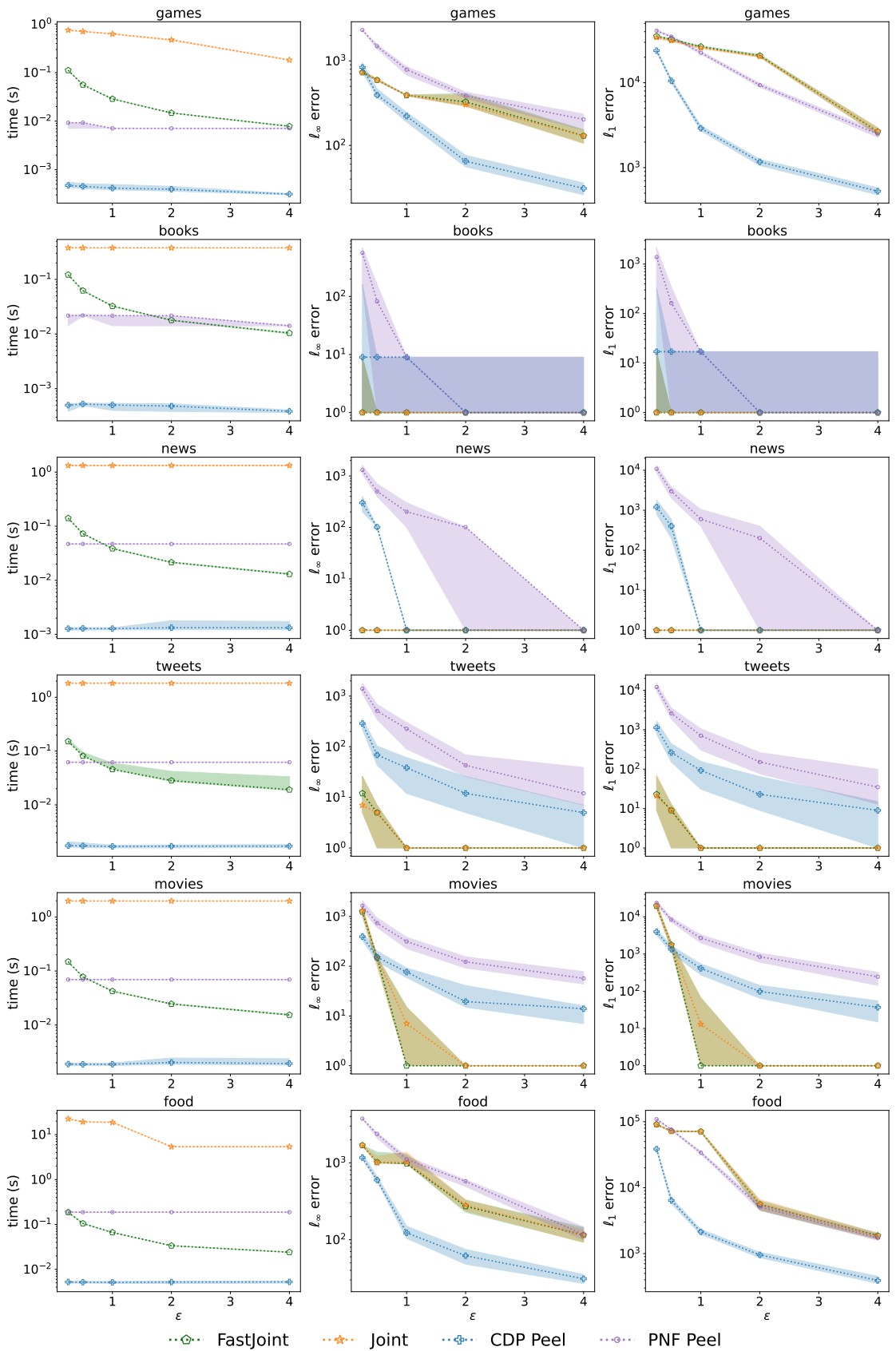

Figure 5: **Left**: Running time vs $\varepsilon$. **Center**: $\ell_\infty$ error vs $\varepsilon$. **Right**: $\ell_1$ error vs $\varepsilon$. The $\ell_1/\ell_\infty$ plots are padded by $1$ to avoid $\log 0$ on the $y$-axis.

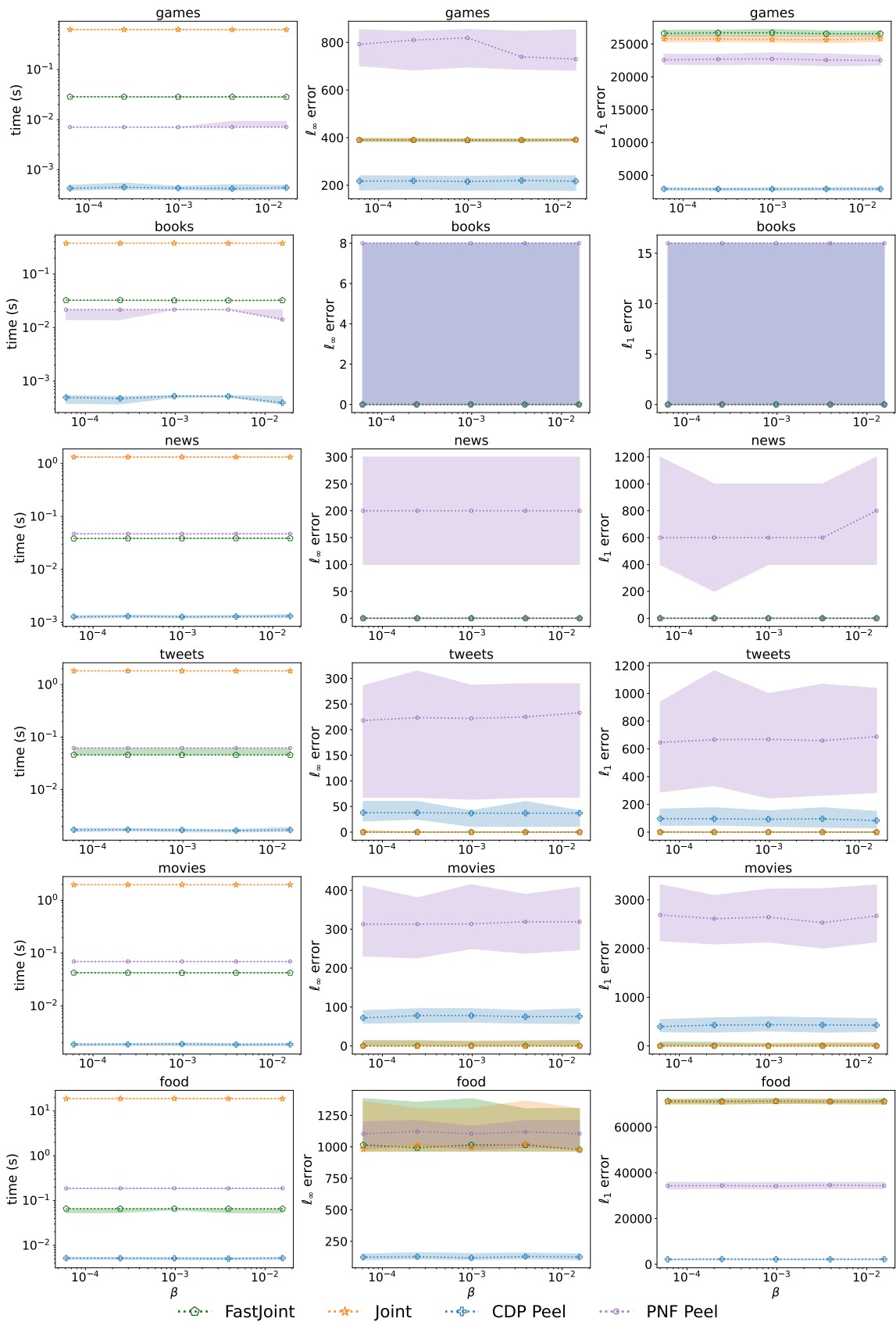

Figure 6: **Left**: Running time vs $\beta$.  **Center**: $\ell_\infty$ error vs $\beta$.  **Right**: $\ell_1$ error vs $\beta$. The $\ell_1/\ell_\infty$ plots are padded by $1$ to avoid $\log 0$ on the $y$-axis.

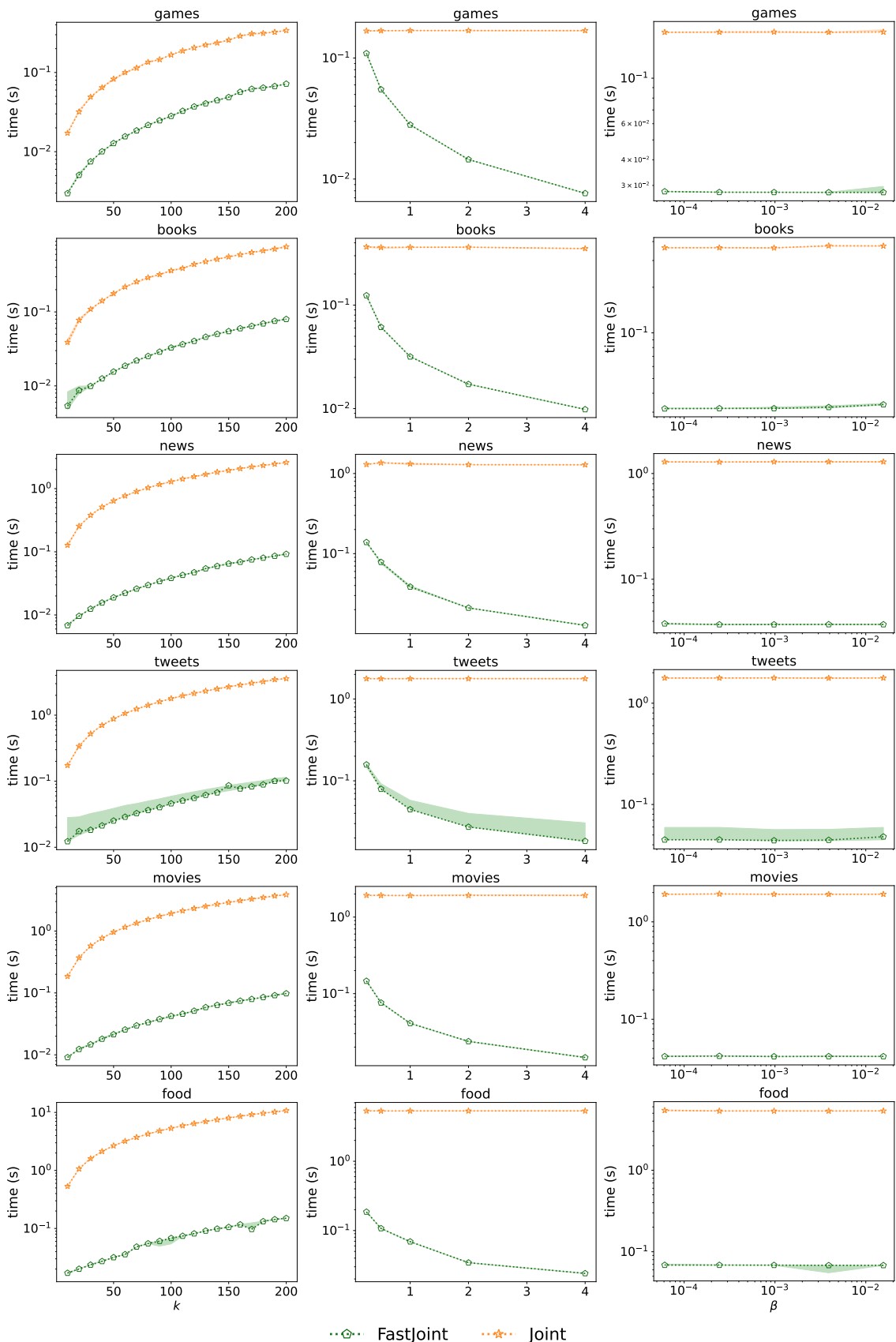

Figure 7: **Left**: Running time vs $k$. **Center**: Running time vs $\varepsilon$. **Right**: Running time vs $\beta$.

