# OpenReview forum: "Faster Differentially Private Top-$k$ Selection: A Joint Exponential Mechanism with Pruning"
_NeurIPS.cc/2024/Conference — NeurIPS 2024 poster_

### Official Review · Reviewer_owmk · 2024-07-08

**Soundness:** 4
**Presentation:** 3
**Contribution:** 2
**Rating:** 6
**Confidence:** 3

**Summary:**

The paper presents a compute and memory efficient implementation of the differentially private top-$k$ algorithm proposed by Gillenwater et al. (2022) which is based on applying the exponential mechanism to all index sequences of length $k$. The implementation by Gillenwater et al. (2022) has the time complexity $O(d k \log k + d \log d )$ and space complexity $O(dk)$, where $d$ is the number of items, whereas this method has the complexity $\widetilde{O}(d + k^2)$, where $\widetilde{O}$ omits the log factors. The experimental comparisons on various datasets verify that this method gives the same utility as the method by Gillenwater et al. (2022) and has 1-2 orders of magnitude smaller running time.

**Strengths:**

- The paper is very well written, in particular the motivation and introduction to the topic
- Clear contribution to an existing method that is in some sense state-of-the-art (has the best accuracy over other baselines in some of the cases)

**Weaknesses:**

- Fairly narrow contribution after all, as it seems that the only contribution is to reduce the memory consumption and compute time of an existing method. The reduction is big but does not reach the speed of the baseline method "CDP Peel" by Rogers et al. (2019) which has also better utility in a large part of the datasets. CDP Peel seems to be the only one of the baseline methods that reaches the speed of the order of milliseconds or less on a standard CPU. And, in most cases, the baseline implementation by Gillenwater et al. (2022) is already quite fast (much less than a second).

- Related to the above comment: there is kind of missing the bigger picture, like when is this implementation useful? When is it useful to use this method vs. CDP Peel?

**Questions:**

Could you answer these questions ( I am ready to reconsider my score ):

When is this impementation really needed? Are you thinking of an online setting? If yes, would the speed be enough for the online setting (considering that the online requirements are often in the order of milliseconds) ? Why not to use CDP Peel if it is so much faster and when is it better?

Minor comments:

- p.1, line 31: "the its"
- p.3, line 96: "lose function"
- p.3, Equation 4: Please add some definition/explanation of $s$
- p.5, line 177: "is decreases"

**Limitations:**

Yes.

---

> ### Author Rebuttal · Authors · 2024-08-06
>
> Thank you for the suggestions. We will provide a clearer overview of the comparison between the approaches in the final version.
>
> 1. When there is a large gap between the $k^{\text{th}}$ and $(k + 1)^{\text{th}}$ largest counts, FastJOINT (our algorithm) and JOINT (by Gillenwater et al. 2022) can achieve smaller empirical error than CDP Peel. This finding is consistent with Gillenwater et al. (2022).
>
> 2. JOINT has $O(dk \log k + d \log d)$ time usage and space usage of $O(dk)$. Even in the offline setting, it suffers from scalability issues. Our approach, FastJOINT, reduces both running time and space to $O(d + \frac{k^2}{\epsilon} \cdot \ln d)$ while providing similar empirical error. It offers a speedup compared to JOINT for free.
>
> 3. JOINT is slow in our experiments. For example, on the food dataset ($d \approx 0.16 \cdot 10^6$, $k = 200$), it took more than 40 seconds. Given this observation, we can anticipate extremely long running times for JOINT on larger datasets, such as those with $d = 10^8$.

---

> > ### Comment · Reviewer_owmk · 2024-08-08
> >
> > Thank you for the replies! I would still have a question: in case you have a large gap between $k$th and $(k+1)$th largest elements, wouldn't it make sense to compare this method to some Propose-Test-Release type of algorithm, where you first test whether there is a large gap between the scores? If you know that there is a large gap, you can release the top-$k$ elements without additional privacy cost. And that would also probably have the complexity $\widetilde{O}(d)$. What would be the benefits of this JOINT approach then?

---

> > > ### Author Response · Authors · 2024-08-09
> > >
> > > Thank you for the insightful question.
> > >
> > > When tackling the version of the top-$k$ selection problem where items must be returned as an ordered sequence and $k$ is fixed, the Propose-Test-Release (PTR) style approach encounters two key challenges:
> > >
> > > 1. **Unordered Output:** PTR may only return the top-$k$ items as an unordered set because two neighboring datasets can have the same collection of top-$k$ elements in a different order. In contrast, the algorithms we compare in the paper return them as an ordered sequence.
> > >
> > > 2. **No Output:** PTR might not return any meaningful result at all (it could return a $\bot$).
> > >
> > > To overcome these challenges, it is essential to use the algorithms discussed in the paper—such as FastJOINT, JOINT, and CDP-Peel—as fundamental tools.
> > > Therefore, studying and comparing the relative performance of these primitives independently is still valuable.
> > > We will incorporate relevant discussions to provide a broader perspective and better position our paper.

---

> > > > ### Author Response · Authors · 2024-08-09
> > > >
> > > > We delve into these challenges in more detail and demonstrate how the algorithms studied in our paper can be leveraged to address them.
> > > >
> > > > - **First Issue: Unordered Output:** This occurs when PTR determines that the gap between the $k^{(th)}$ and $(k + 1)^{(th)}$ largest elements is sufficiently large, resulting in the top-$k$ items being returned as an unordered set. To resolve this, one can use Joint-style algorithms, which are effective when there are still large gaps between the $i^{(th)}$ and $(i + 1)^{(th)}$ largest elements for $i \leq k$.
> > > >
> > > > - **Second Issue: No Output:** This happens when PTR concludes that the gap between the $k^{(th)}$ and $(k + 1)^{(th)}$ largest elements is not significant enough. We can consider two potential solutions:
> > > >
> > > >   1. **Invoke a Private Top-$k$ Selection Algorithm:** Use one of the private top-$k$ selection algorithms discussed in the paper. JOINT-style algorithms are particularly suitable if there are still large gaps between the $i^{(th)}$ and $(i + 1)^{(th)}$ largest elements for $i = 1, 2, ..., k - 1$. In this case, JOINT can reliably identify the first $k - 1$ largest items, allocating the privacy budget  primarily on the last item. In contrast, CDP-Peel would need to split the privacy budget across all $k$ items.
> > > >
> > > >   2. **Use Report Noisy Max:** Use the Report-Noisy-Max technique to approximately identify the index $i$ that maximizes the gap between the $i^{(th)}$ and $(i + 1)^{(th)}$ largest elements. After identifying this index, test whether the gap is sufficient to determine the top-$i$ set. However, if $i < k$ —- which occurs when the gap between the $i^{(th)}$ and $(i + 1)^{(th)}$ largest elements is significantly larger than the gap between the $k^{(th)}$ and $(k + 1)^{(th)}$ —- JOINT-style algorithms can be employed to identify the remaining top-$(k - i)$ items, assuming there are still large gaps between their counts.
> > > >
> > > >         This approach draws inspiration from Zhu and Wang ("Adaptive private-k-selection with adaptive k and application to multi-label pate", 2022), where they considered using Report-Noisy-Max alongside CDP-Peel, and focused on returning the top-$k$ items as an unordered set.

---

> > > > > ### Comment · Reviewer_owmk · 2024-08-09
> > > > >
> > > > > Thank you for the replies! This makes it clearer and I will raise my score.
> > > > >
> > > > > > We will incorporate relevant discussions to provide a broader perspective and better position our paper.
> > > > >
> > > > > Please do so! I think this would be very valuable, based on a quick glance it seems that the original JOINT paper is not making these connections.

---

> > > > > > ### Author Response · Authors · 2024-08-11
> > > > > >
> > > > > > Thank you for the suggestion and for appreciating our work. We will include the relevant discussions in the final version.

---

### Official Review · Reviewer_pna3 · 2024-07-13

**Soundness:** 3
**Presentation:** 3
**Contribution:** 3
**Rating:** 6
**Confidence:** 3

**Summary:**

This paper considers a core differentially private primitive of top-k selection. In this problem, there are $d$ items where each item gets a frequency $h[i]$ which is the sum of $n$ binary vectors $\{0,1\}^d$. The goal is top select (approximate) top-k most frequent items while preserving privacy with respect to the addition or removal of a single vector (corresponding to a user's contribution).

The paper begins by calling attention to a recent work of Gillenwater et al. which achieves good empirical performance for this problem via the exponential mechanism using an algorithm called Joint. The algorithm assigns a score to every possible $\binom{d}{k}$ subset. In order to efficiently implement sampling proportionally to the exponentiation of the score, the Joint algorithm considers different level sets of subsets which can be efficiently sampled from. The total running time of this algorithm is $\tilde{O}(dk)$.

In this work, the authors give a faster algorithm that implements (essentially) the same exponential mechanism algorithm that runs in time $\tilde{O}(d + k^2/\epsilon)$. The main idea of the algorithm is to truncate the score function to be at most a certain value which allows for faster computation without worsening the quality of the algorithm by much.

**Strengths:**

- The problem is well-motivated and important within differential privacy.
- The proposed change to prior work is simple and yields a significant benefit in the theoretical bounds.
- The experiments are detailed and show across the board that the new algorithm is much faster than Joint while retaining its performance, which is often better than other baselines.

**Weaknesses:**

- It is not entirely clear to me what part of section 4 is novel in this paper and what part comes from the prior work of Gillenwater et al. See my questions below. Is the setup of subsets from which to sample different from Gillenwater in addition to the loss function? Or is it only truncating the loss function that yields the benefits in the paper?
- Include $\epsilon$ in the running time. Especially as it only appears for the final version of your algorithm and not elsewhere, it is important when comparing runtimes.
- Minor: Order the references alphabetically.

**Questions:**

- Is the $O(d + nk)$ time version of the algorithm before pruning already new in this work?

---

> ### Author Rebuttal · Authors · 2024-08-06
>
> Thank you for the suggestion. We will order the reference list alphabetically in the final version, and include the $\varepsilon$ in the running time.
>
> We will also clarify our contribution at the beginning of Section 4 in the final version:
>
> 1. Section 4.1 is novel. It includes
>
>    * a new "group by" framework for sampling a sequence from the space comprising all $d^{\Theta(k)}$ possible length-$k$ sequences directly;
>
>    * a new partition for the sampling space, which leads to the $O(d + nk)$-time algorithm; and
>
>
>    * a new pruning technique (by truncating the loss function), which reduces the running time to $O(d + \frac{k}{\varepsilon} \cdot (k\ln d + \ln\frac{1}{\beta}))$.
>
>
> 3. Section 4.2 compares our approach with the one by Gillenwater et al. To facilitate the comparison, we explain Gillenwater et al's algorithm using our newly proposed framework. Note that they applied a different partition of the sampling space, and our pruning technique does not work for their partition.

---

> > ### Comment · Reviewer_pna3 · 2024-08-07
> > **Response to rebuttal**
> >
> > Thanks for addressing my question. I will raise my score.

---

### Official Review · Reviewer_3vYL · 2024-07-14

**Soundness:** 4
**Presentation:** 4
**Contribution:** 3
**Rating:** 7
**Confidence:** 4

**Summary:**

This paper studies differentially private top-k selection i.e., selecting the top k sums from a set of d different ones. This problem has been well studied and mechanisms achieving asymptotically optimal error are known. Durfee and Rogers [2019] gives an optimal mechanism that takes O(d) time. However, in 2022, Gillenwater et al. proposed an alternate mechanism based on the exponential mechanism directly run on the set of all sequences of length k, and showed that it achieved better empirical performance on many datasets. While naive implementations can take exponential time, they showed how this algorithm can be implemented in time O(dk).

In this paper, the primary focus is on improving the computational complexity of the algorithm described by GIllenwater et al. to $O(d + k^2)$ which matches the Durfee et al. time complexity for most $k$ of practical interest.

Their general framework involves partitioning the sequence space (such that every subset in the partition has identical quality), and performing a two step sampling- firstly, sampling a subset in the partition as per the exponential mechanism and then outputting a uniformly random element in the subset. The partition consists of subsets $S_{r,i}$ ($r$ denotes the error and $i$ a coordinate) consisting of sequences where the first coordinate achieving the error $r$ is $i$. They show that using a recursive approach, with $O(d)$ preprocessing time, the subset sampling can be performed in $O(nk)$ time and the uniform sequence sampling in $O(d)$ time. Finally, since with high probability the exponential mechanism does not sample a sequence with error larger than $O(k)$, by truncating the score function used in the exponential mechanism (i.e. max error) to never be larger than $k$, the time complexity of the algorithm can be reduced to $O(d+k^2)$. They also back up their theoretical results with experimental work confirming their findings.

**Strengths:**

1) Top-k selection is a fundamental problem that abstracts many important optimization tasks. Hence, more efficient implementations of the algorithms achieving the best empirical accuracy for this problem are valuable contributions.

2) Improving the time complexity of the algorithm requires a novel partitioning approach. As argued by the authors, fitting the previous Gillenwater et al. approach into the same partitioning framework does not allow for the time complexity improvement that comes from truncating the score function.

3) The paper is well written and thorough and the experimental work explains clearly where we can expect the improved FastJoint algorithm to be better than other algorithms described for this problem.

**Weaknesses:**

The paper doesn't have any weaknesses that I can pinpoint.

**Questions:**

I had a couple of questions about the experiments:

1) On the movies dataset, the FastJoint algorithm shows larger variance than other methods (and also more than FastJoint on other datasets). I'm curious why this might be.

2) For the food dataset, the sudden drop in variance of the error of FastJoint close to k=100 seems very strange- what happened here?

3) For small datasets, is there any hope to utilize the benefits of vectorization for FastJoint as well? Some comment may be nice.

**Limitations:**

Yes.

---

> ### Author Rebuttal · Authors · 2024-08-06
>
> Thank you for being supportive of our manuscript.
>
>
>
> 1. On the movie dataset, the JOINT (orange color) has a larger variance than FastJOINT (green color).
> In this dataset, there is a visually noticeable variation in the error plot around $k = 100$ (Figure 1).
> However, this variation is actually small, ranging between $10^0$ and $10^1$ on the $y$-axis.
>
>
>
>
> 2. We check the original experimental data for the food dataset. The loss does not drop after $k \ge 100$; it remains roughly the same. Indeed, if we create a screenshot of Figure 1 for the food dataset and draw a horizontal line near the $\ell_\infty$ error plot of FastJoint, we can see that the error plot of FastJoint is parallel to the horizontal line.
>
>
>     The reason that the error remains stable can be related to the fact that the gap between the $k^{\text{th}}$ and $(k + 1)^{\text{th}}$ largest counts becomes small and stable after $k \ge 100$ (which can be seen from the plots of the gaps in Figure 4 on page 16 in the appendix), and that FastJoint's error is gap-dependent.
>
>
>
> 3. The only $O(d)$ time part of the algorithm is computing the histogram of the score, which can be easily vectorized.
> While the rest of FastJoint is hard to vectorize, it is not the time bottleneck when running on an industrial dataset (e.g., $d = 10^8$, $k = 100$).

---

> > ### Comment · Reviewer_3vYL · 2024-08-09
> >
> > Re 3, I was referring to the discussion starting line 268 of the paper regarding PNF peel being faster than FastJoint on small datasets (where I'm assuming $d$ is smaller) because of vectorization. It's good to know however that the $O(d)$ part can be vectorized.
> >
> > Re 1, I was referencing the time plot (not the error plot).
> >
> > Re 2, I was referring to the sudden drop in variance of error for the food dataset around k=90 (see some of the area shaded only Orange) . FastJoint's error is indeed gap dependent, but should the same phenomenon not apply then to Joint as well? Additionally, this is only around k=90 but after that the variance goes up again, so it seems curious.

---

### Official Review · Reviewer_u5C6 · 2024-07-16

**Soundness:** 3
**Presentation:** 3
**Contribution:** 3
**Rating:** 7
**Confidence:** 3

**Summary:**

This paper gives a faster method of performing differentially private top k selection. Previous work by Gillenwater et al., introduced the joint exponential mechanism in the context of top k selection and gave a DP algorithm that used \tilde{O}(kd) time and space. The main contribution of this paper is a new sampling technique which uses a variant of the joint exponential mechanism and improves the time and space to \tilde{O}(d+k^2).

**Strengths:**

The problem addressed by the paper is an important one, and the results are novel and improve upon the state of the art. Overall I think the paper does a decent job of comparing and contrasting their techniques with that of the previous work. The subsampling and pruning techniques are quite effective and have been described adequately well in the main body. The experiments support claims made in the paper.

**Weaknesses:**

Since the paper builds on the Joint exponential mechanism from previous work, it would be helpful to have a more detailed supplementary section about this in the appendix for e.g. to give more background.

**Questions:**

I don't have any specific questions, just a suggestion to add more background on the Joint Exponential Mechanism in the appendix since you are building on this framework.

---

> ### Author Rebuttal · Authors · 2024-08-06
>
> Thank you for the suggestion.
> We will add more background on the Joint Exponential Mechanism in the appendix.
> Specifically, we will include brief explanations and proofs for the properties of the Joint Exponential Mechanism (Facts 3.5 and 3.6) to ensure completeness.

---

> > ### Comment · Reviewer_u5C6 · 2024-08-09
> >
> > Thanks for addressing my suggestions.

---

### Author Rebuttal · Authors · 2024-08-06

We thank all the reviewers for their consideration of our manuscript and their constructive feedback.

---

### Decision · Program_Chairs · 2024-09-25

**Decision:**

Accept (poster)

**Comment:**

The reviewers remark that the manuscript studies an important/fundamental problem [u5C6, pna3, 3vYL], makes a novel/non-trivial contribution [u5C6, 3vYL, owmk, pna3] which is a clear improvement upon the state-of-the-art [u5C6, owmk], explanations are clear / paper well-written [u5C6, 3vYL, owmk] and experiments substantiate claims [u5C6, 3vYL, pna3], but it could be clearer which parts can be implemented with vector-wise operations [3vYL], which sections are novel [pna3], it would be useful to provide a bit more background on the joint exponential mechanism [u5C6] and considering that the main contribution are improvements in sampling speed & memory usage, how big the advantage is over CDP peel that is even more efficient [owmk].

The rebuttal answers reviewer questions, agrees to add more background on the joint exponential mechanism [response to u5C6], clarifies that section 4.1 is novel [response to pna3] and which parts can be vectorised [response to 3vYL] and clarifies advantages over CDP peel and contextualises it within the propose-test-release approach [response to owmk].

The overall recommendation is to accept the submission with the changes agreed upon in the discussion phase as the proposed approach mitigates one major drawback of the joint exponential mechanism, i.e., sampling efficiency, without sacrificing utility, offers good insights and ideas relevant to the problem and the manuscript substantiates claims appropriately.